

# Real-time organic aerosol characterization via Orbitrap mass spectrometry in urban and agricultural environments

Julia David[1], Luca D'Angelo[1], Mario Simon[1], Alexander L. Vogel[1]

[1] Institute for Atmospheric and Environmental Sciences, Goethe University Frankfurt, 60438 Frankfurt am Main, Germany

*Correspondence to*: Julia David (david@iau.uni-frankfurt.de), Alexander Lucas Vogel (vogel@iau.uni-frankfurt.de)

**Abstract**

Mass spectrometry techniques traditionally deployed in the field often operate at low mass resolution, making it hard to unambiguously identify and attribute organic molecules. In this regard, in-situ, accurate and precise online mass-spectrometric measurements of organic molecules in atmospheric organic aerosol (OA) are essential for understanding its sources, formation

and chemical composition. In this study, we demonstrate the field applicability of a high-resolution (Orbitrap) mass spectrometer with Atmospheric Pressure Chemical Ionization (APCI-Orbitrap-MS) for real-time ambient OA measurements, achieving online, molecular resolution at atmospherically relevant concentrations with a high temporal resolution of 1 s, mass resolution of R= 120,000 at $m/z$ 200, and mass accuracy of ±1.5 ppm. These features enable chemically reliable measurements in environments that are exhibiting chemically complex aerosol composition, through molecular-level detection and

identification of anthropogenic pollutants, biogenic and biomass burning tracers. As proof of principle, we deployed the APCI-Orbitrap-MS for in-situ measurements in a mobile laboratory container at an urban background station at Campus Riedberg (CR, Frankfurt am Main, Germany) and an agricultural field site in Schivenoglia (SKI, Italy) in the heavily polluted Po Valley. The APCI-Orbitrap-MS showed good agreement with the organic aerosol mass of an aerosol chemical speciation monitor (ACSM), with Pearson's R values of 0.91 and 0.70 for the urban and agricultural sites, respectively. In SKI, we resolved

distinct diurnal variations in compounds such as MBTCA ($C_8H_{12}O_6$), a biogenic marker of photochemical aging, and $C_8H_{13}O_8N$, an organic nitrate indicative of nighttime chemistry. Additionally, nighttime biomass burning events were detected frequently, with durations ranging from 10 to 40 minutes, emphasizing the importance of high temporal resolution. During these events we found up to 30 isobaric peaks per unit mass that are baseline-resolved. For the first time, the hydroxypinonyl ester of cis-pinic acid ($C_{19}H_{28}O_7$) could be measured and confirmed with $MS^2$ experiments in ambient aerosol by an in-situ

method at CR. In addition, laboratory experiments were performed to confirm the broad applicability of the APCI-Orbitrap-MS for the real-time detection of biogenic and biomass burning tracers, as well as specific anthropogenic pollutants, such as pesticides, organophosphates or organic esters from aircraft lubrication oil.




## 1 Introduction

Atmospheric organic aerosols (OA) are complex mixtures of both, compounds emitted directly and those formed in-situ via (photo)oxidation, oligomerization, and fragmentation. OA is a major component of atmospheric particulate matter (PM),

accounting for 20–50 % of global aerosol loading on average and up to 90 % of $PM_{2.5}$ (PM < 2.5 µm) during severe urban pollution events (Srivastava et al., 2022; Zhang et al., 2024), playing a crucial role in shaping Earth's climate and atmospheric processes through both direct and indirect interactions (Zhang et al., 2007; Intergovernmental Panel on Climate Change, 2023). Related to human health, PM can penetrate deep into the respiratory system, leading to respiratory and cardiovascular diseases (Pope and Dockery, 2006). It is estimated that globally 8.8 million premature deaths occur annually due to exposure to air

pollution and elevated levels of $PM_{2.5}$ (Lelieveld et al., 2020; World Health Organization, 2021). While adverse health and climate effects of aerosols are recognized, there remains a lack of knowledge about their chemical composition. Understanding aerosols' composition is crucial for assessing aerosol properties related to health, such as reactivity or toxicity, as well as those related to climate, such as hygroscopicity or light absorption. An individual atmospheric aerosol particle can contain hundreds to thousands of different organic compounds, often in very low concentrations (Goldstein and Galbally, 2007). Their reactivity

contributes to detrimental health effects through the formation of radicals or reactive oxygen species (Pope and Dockery, 2006; Park et al., 2018). Despite the low concentrations of specific organic molecules, their detection enables the identification of the sources and processes contributing to air pollution. However, aerosols undergo constant transformation and degradation, and are removed from the atmosphere by wet and dry deposition, which is why local aerosol pollution is difficult to model and globally aerosols are termed short-lived climate forcers (Volkamer et al., 2006; Couvidat et al., 2018; Szopa et al., 2021). To

capture the high variability of OA composition and its implications on both climate and health, it is therefore essential to employ instrumentation that provides reliable and comprehensive chemical composition measurements at high temporal resolution (Zahardis et al., 2011; Nozière et al., 2015).

Mass spectrometry (MS) is the most employed technique to resolve complex composition and chemical processes (Johnston and Wexler, 1995; Hallquist et al., 2009; Nozière et al., 2015). Offline methods involve collecting PM on filter material or in

liquids, followed by sample storage, sample extraction (e.g. liquid extraction, solid phase extraction, thermal desorption, ultrasonication) and pretreatment (e.g. derivatization) before analysis using chromatographic separation coupled with MS (Hallquist et al., 2009; Doussin et al., 2023). These methods enable detailed chemical characterization, sample enrichment and chromatographic separation, with the added benefit of archiving samples for future analysis (Zuth et al., 2018; Resch et al., 2023). However, offline techniques suffer from low time resolution (hours to days) and are prone to artifacts during collection,

storage, and processing, such as reactions, degradation, or evaporation of compounds, which can distort the representativity of the measured PM (Vogel et al., 2013; Zuth et al., 2018; Lopez-Hilfiker et al., 2019; Resch et al., 2023; Resch et al., 2024).

Alternatively, online measurement techniques have been developed to measure the chemical composition of the gas and particle phase separately or simultaneously. These techniques vary in their ionization methods and time resolution, ranging from semi-online techniques with resolution of minutes to hours to real-time measurements with a resolution of seconds. The

aerosol mass spectrometer (AMS) and aerosol chemical speciation monitor (ACSM) are well-established devices for measuring ambient aerosol using electron ionization (EI), a hard ionization technique. These techniques quantify chemical species of ambient aerosols, including organics, sulfate, nitrate, ammonia, and chloride. Nevertheless, this method is incapable of providing individual molecular information due to the strong fragmentation that occurs using EI (Jimenez et al., 2003; Nash et al., 2006; Allan et al., 2004; Allan et al., 2003). When combined with positive matrix factorization (PMF), it is however

possible to identify factors associated with diverse organic compositions and to allocate potential aerosol sources (Ulbrich et al., 2009; Koss et al., 2020). Additionally, specific fragments, such as $f_{44}$ (*m/z* 44, $CO_2^+$) and $f_{43}$ (*m/z* 43, $C_2H_3O^+$ and $C_4H_7^+$),



can provide insights into photochemical aging and the oxidation state of organic aerosols (Ng et al., 2010; Ng et al., 2011a; Lambe et al., 2011).

To get a better understanding of the molecular composition of OA, soft ionization techniques that keep molecules to a certain extent intact, have been developed in the past decades. A widely used soft ionization particle measurement technique is the combination of chemical ionization with the filter inlet for gases and aerosols (FIGAERO), allowing to measure online gas-phase composition whilst collecting particles on a filter. The OA composition and volatility is subsequently measured by thermal desorption of the filter. Switching between gas and particle phase measurement comes at the expense of a high time
resolution (Lopez-Hilfiker et al., 2014; Thornton et al., 2020). The chemical analysis of aerosol online (CHARON) inlet system eliminates this downside and enables real-time analysis by directly sampling PM without prior collection. When coupled with a proton transfer reaction time-of-flight MS (PTR-ToF-MS), it quantitatively measures organic analytes and ammonia but can exhibit strong fragmentation during PTR ionization (Eichler et al., 2015; Peng et al., 2023). The vaporization Inlet for Aerosols (VIA) consists of an activated charcoal denuder to remove gas-phase compounds and a vaporizer tube to thermally vaporize
aerosol (Voisin et al., 2003; Zhao et al., 2024). It is typically coupled to a chemical ionization mass spectrometer (CIMS), allowing the selective detection of oxidized organic species. VIA's design is effective for highly oxidized species but can cause thermal decomposition already at 250 °C and operates depending on the coupling at non-atmospheric pressure, resulting in additional fragmentation and lower ionization efficiencies (Johnston and Kerecman, 2019; Zhao et al., 2024). Further soft-ionization techniques are the extractive electrospray ionization (EESI) and the aerosol flowing atmospheric-pressure afterglow
(AeroFAPA). EESI uses charged droplets generated by an electrospray probe to ionize soluble components of PM, which are extracted and subsequently ionized during droplet evaporation. The use of solvents however can introduce artifacts (Lopez-Hilfiker et al., 2019; Lee et al., 2020). AeroFAPA, employs helium glow discharge plasma to create reagent ions for analyte ionization (Shelley et al., 2011; Brüggemann et al., 2015).

To overcome insufficient time resolution while minimizing decomposition, fragmentation and solvent artifacts, we utilized an
Atmospheric Pressure Chemical Ionization (APCI) ion source for this study. The applicability of APCI for laboratory based online measurements has been demonstrated in multiple studies (Hoffmann et al., 1998; Kückelmann et al., 2000; Vogel et al., 2013; Zuth et al., 2018). The APCI can be operated in either positive $[M+H]^+$ or negative $[M–H]^-$ ionization mode. When using APCI, the positive ionization mode is suitable for molecules with functional groups that have lone electron pairs, such as carbonyls, amines, esters, alcohols and organic hydroperoxides. In contrast, molecules like organic acids, alcohols and
multifunctional organic compounds, including peroxy-acids, are preferably ionized in negative mode due to their high gas-phase acidity (Hoffmann et al., 1998; Zahardis et al., 2011; Zhou et al., 2018). Combining APCI with high resolution MS, such as the Orbitrap-MS, enables the unambiguous attribution of molecular formulas to individual peaks in the spectra (Hernández et al., 2012; Zuth et al., 2018; Cai et al., 2021; Yuan et al., 2024).High resolution not only minimizes peak overlap but also enables the identification of isotopic variants and isobaric species, reducing uncertainty in molecular formula assignment even
further (Hernández et al., 2012; Nagao et al., 2014).

In this paper we demonstrate the applicability and performance of APCI coupled to an Orbitrap-MS for in-situ measurements. Continuous operation inside a mobile laboratory container (MLC) at an urban background and agricultural field site was achieved, and its performance is compared against the ACSM. We show that high mass resolution (R= 120 000 at *m/z* 200) and high mass accuracy (± 1.5 ppm) is necessary for the molecular level identification of compounds in such complex ambient
environments. We further characterize and validate our findings by laboratory experiments including nebulized standards and an oxidation flow reactor (OFR) to characterize aerosol losses, ionization efficiencies and the instrumental sensitivity for anthropogenic, biogenic, and biomass-burning (BB) compounds. Comparing fragmentation spectra (MS[2]) of reference standards with field-acquired fragmentation spectra demonstrates the possibility of improved compound identification in real-time.



## 2 Experimental Section

### 2.1 Instrumental setup

The online measurements of OA were performed using a high-resolution Orbitrap Exploris 120 mass spectrometer (Thermo Fisher Scientific Inc.), acquiring spectra with a resolution of 120,000 at $m/z$ 200. For ionization, we use APCI as it minimizes analyte fragmentation and enables molecular ion measurement. For this, the original OptaMax NG (Thermo Fisher Scientific Inc.) APCI ion source was modified as described by Hoffmann et al. (1998) and Kückelmann et al. (2000) by removing the capillary from the APCI probe and opening the nozzle mechanically to a diameter of 2.4 mm. This modification enables a higher sample flow into the ion source. At the same time the sheath, auxiliary, and sweep gases were switched off to prevent dilution of the aerosol sample.

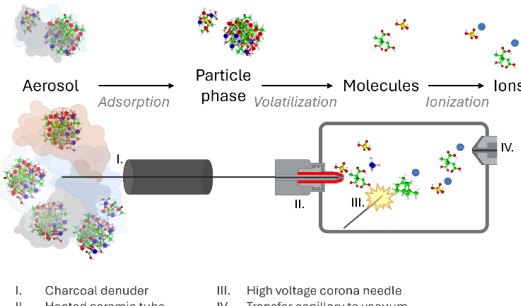

**Figure 1: APCI-Orbitrap-MS schematic visualizing the process from aerosol to particle phase to molecules and ions. Dimensions are not true to scale.**

Prior to the ion source, gas-phase is removed by a honeycomb multichannel denuder (length: 10 cm, outer diameter: 3.5 cm, ~ 600 channels). The particle phase is then vaporized in a heated ceramic tube (set point 350 °C (see section 3.1)), resulting in gas-phase compounds that are subsequently ionized by corona discharge and transferred into the vacuum system of the Orbitrap-MS (Fig. 1). In negative ionization mode, $O_2^-$ is formed under atmospheric pressure by the corona discharge between the corona needle and the transfer capillary. Gaseous analytes, which have a higher gas-phase acidity than $O_2^-$ become ionized by charge transfer and/or proton transfer reactions. The remaining excess energy of the proton transfer reaction is transferred by collision to a third neutral molecule. For the positive ionization mode, positively charged analyte ions are formed by the proton transfer reaction from $H_3O^+$ to the analyte. $H_3O^+$ is formed in the ion source by a chain of reactions where $N_2$ is oxidized and subsequently forming $H_3O^+$, which in turn transfers a proton onto the analyte molecule (Carroll et al., 1975; Sharon and Bartmess).

In order to achieve both a high time resolution and sensitivity, a complete spectrum was recorded at one-second intervals ranging from $m/z$ 85 to $m/z$ 650 (Tab. 1). The lower mass threshold was set to exclude small compounds with a high abundance (e.g. nitric acid) and thus ensure that the ion trap is not unnecessarily filled with these ions. This scan window also has the advantage of increasing the sensitivity to larger organic compounds (Cai et al., 2021). To ensure high mass accuracy, the APCI-Orbitrap-MS was calibrated once a week using the Thermo Scientific Flexmix (CAS: 75-05-8). Furthermore, it was continuously operated with a lock mass calibration of $m/z$ 255 (palmitic acid) and an internal calibration of $m/z$ 203 (fluoranthene) in negative and positive ionization mode, respectively. For background measurements, the sample flow was drawn through a high-efficiency particulate air (HEPA) filter for up to 30 min.





145       **Table 1: Instrumental settings for online APCI-Orbitrap-MS measurements in negative and positive ionization mode.**

|  | Negative ionization mode | Positive ionization mode |
|---|---|---|
| Vaporizer temperature (°C) | 350 | 350 |
| Ion transfer tube temperature (°C) | 200 | 200 |
| Ion source gas flows (a.u.): sheath-, aux-, sweep gas | 0[*] | 0[*] |
| Discharge current (µA) | 3 | 3 |
| Scan range ($m/z$) | 80–650 | 80–650 |
| RF lens (%) | 50 | 50 |
| AGC Target | Standard | Standard |
| Internal mass calibration ($m/z$) | 255.2329 | 203.0855[†] |
| Injection time (ms) | 100 | 100 |
| Averaged microscans | 10 | 10 |

[*]Sheath-, aux- and sweep gas were 0 a.u., due to the disconnection of $N_2$ supply.

[†]Internal calibration using EASY-IC™

To test the in-situ applicability of the APCI-Orbitrap-MS, we installed it inside a MLC, in which all particle measurement devices (Fig. S1 and S2) were connected to a $PM_{2.5}$ inlet system with a laminar flow splitter. The sample flow into the APCI was maintained at 1.6 L min$^{-1}$, by connecting an external pump to the exhaust of the Orbitrap Exploris 120. To ensure

measurement of only the particle-phase, the honeycomb-multichannel active charcoal denuder effectively removed gas-phase compounds while allowing particle-phase transmission. At a sample flow rate of 1.6 L min$^{-1}$, the denuder achieved a 100 % removal efficiency for α-pinene at approximately 600 ppb (Fig. S3). Particle losses were determined to be 19.23 % based on a polydisperse particle size distribution (Fig. S4).

To validate the performance of the field measurements of the APCI-Orbitrap-MS, we measured in parallel with a ToF-ACSM

(Aerodyne Research Inc.), which determines the mass concentration of the main non-refractory species in sub-micrometer particles fraction (near-$PM_1$), namely organics, nitrate, sulfates, ammonium and chloride (Fröhlich et al., 2013). This is obtained by thermal particle vaporization at 600 °C in a standard vaporizer chamber, followed by EI ionization at 70 eV and detection by a ToF mass spectrometer (Ng et al., 2011b). Voltage tuning was performed after installation of the device and before operating the relative ionization (RIE) calibration of ammonium and sulfate. For this purpose, a pure solution of

ammonium nitrate and ammonium sulfate was nebulized, and the ion count of the device was measured in accordance with the calibration method proposed by the manufacturer and the standard operating procedures of COLOSSAL/ACTRIS (version 1, March 2022). The instrument was equipped with a multichannel Nafion aerosol dryer (Aerodyne Research Inc.) using a counterflow of 3 L min$^{-1}$.

### 2.2 Laboratory experiments

To characterize the APCI-Orbitrap-MS we generated biogenic OA by oxidation of α-pinene (98 %, Alfa Aesar, CAS: 7785-26-4) using a potential aerosol mass oxidation flow reactor (PAM-OFR, Aerodyne Research Inc.). The experiments followed the protocols of Lambe et al. (2011) and Thoma et al. (2022). We set the overall flow through the PAM-OFR to 4.5 L min$^{-1}$, resulting in an average residence time of three minutes. We introduced externally generated $O_3$ to simulate dark ozonolysis conditions without OH scavenger. The relative humidity in both experiments ranged from 40–60 %. In Experiment I, we

injected approximately 600 ppb of α-pinene and around 2 ppm of $O_3$, which resulted in SOA mass concentrations between 12 and 35 µg m$^{-3}$. In Experiment II, the $O_3$ concentrations were increased to approximately 5 ppm while α-pinene was elevated to 6,000 ppb, resulting in a mean SOA mass concentrations of approximately 300 µg m$^{-3}$. We recorded the resulting particle formation using a scanning-mobility particle sizer (SMPS, TSI, model: 3938) with a soft X-ray neutralizer.





To determine the sensitivity of the APCI-Orbitrap-MS and perform online MS$^2$, we generated a variety of aerosols with a

nebulizer (replica of TSI model 3076) with different biogenic and anthropogenic compounds dissolved in methanol and water. As reference standards for biogenic compounds, we selected 3-methyl-1,2,3-butanetricarboxylic acid (MBTCA, self-synthesized), and pinic acid (self-synthesized). As BB compounds 1,6-anhydro-β-D-glucopyranose (Levoglucosan, 99%, Alfa Aesar, CAS: 498-07-7), vanillin (99%, Merck, CAS: 121-33-5) and 4-nitrocatechol (97%, Sigma Aldrich, CAS: 3316-09-4) were chosen. As anthropogenic compounds, we selected acridine (98%, Alfa Aesar, CAS: 260-94-6), camphorsulfonic acid

(98%, Aldrich, CAS: 5872-08-2), [(phosphonomethyl)amino]acetic acid with 2-propanamine (1:1) (Glyphosate, 95%, Sigma Aldrich, CAS: 38641-94-0), and Mobil Jet™ Oil II (ExxonMobil Corporation). Downstream of the nebulizer, we installed a silica gel and charcoal denuder to reduce the water and methanol content in the sample flow, respectively. Subsequentially, the chemical composition of the aerosol was analyzed using APCI-Orbitrap-MS, while the particle-size distribution was measured by the SMPS.


### 2.3 Field measurements: Campus Riedberg (Frankfurt, Germany) and Schivenoglia (Po-Valley, Italy)

The MLC was positioned at the meteorological measurement site at Campus Riedberg (CR, 50°10'23.1" N 8°38'05.7" E) to evaluate the chemical composition of aerosols and size-distribution at an urban background location during the summer of 2023. Afterwards, we moved the MLC container to Schivenoglia (SKI, 45°01'00.8" N 11°04'33.7" E), an area of agricultural

activity within the Po Valley, Italy. This site was selected to conduct measurements during the late summer and early fall transition period. In this paper, we will focus on the intensive measurement periods at CR from 07 June 2023 to 10 June 2023 and at SKI from 28 September 2023 to 05 October 2023.

We equipped the MLC for the particle measurement with a PM$_{2.5}$ particle inlet (Digitel Elektronik AG) which was installed vertically 2 m above the container roof. The APCI-Orbitrap-MS, an ACSM and an SMPS were connected to a self-built

stainless steel laminar flow splitter through stainless steel and conductive Polytetrafluoroethylene (PTFE) tubing. Although all particle instruments were connected to the same inlet system, they had different cut-off sizes: the APCI-Orbitrap-MS had a PM$_{2.5}$ cut-off, while the ACSM and SMPS operate at a smaller sub-micron area. The SMPS was equipped with a D50 710 nm impactor, whereas the ACSM measures in an area between 70 and 700 nm, caused here by transmission range of the aerodynamic lenses (Liu et al., 2007). Despite these differences, the comparison remains valid, as the average particle number

distribution peaked at approximately 65 nm at CR and 70 nm at SKI, while the average particle mass distribution peaked around 190 nm at CR (Fig. S5) and 325 nm at (Fig. S6).

### 2.4 Data processing

High-resolution MS data from real-time measurements were processed using Orbitool 2.5.2 beta (Cai et al., 2021), a software tool specifically developed for the analysis of time-series data acquired with the Orbitrap. We denoised and averaged the

spectra over two minutes with a mass tolerance of ±1.5 ppm. The molecular formulas were calculated using: #C (0−30), #H (0−60), #O (0−15), #N (0−10), #S (0−10), #P(0−10). For further interpretation of fragmentation experiments was facilitated by the utilization of FreeStyle™ 1.8 SP1 (Thermo Fisher Scientific Inc.). To describe the level of certainty with which compounds were identified, we used the confidence levels developed for high-resolution MS by Schymanski et al. (2014). Exact mass (level 5) and unequivocal molecular formula (level 4) are always achieved during ambient measurements using

APCI-Orbitrap-MS. Tentative candidates (level 3) and probable structure (Level 2) were achieved with additional laboratory fragmentation experiments (Schymanski et al., 2014). We like to emphasize, that using online mass spectrometry, level 1 cannot be achieved as it is missing the chromatographic separation dimension.

The analysis of the ToF-ACSM data was conducted using Tofware (version 4.0.0), which was operated within the framework of Wavemetrics Igor Pro (version 7). The default values for the ionization efficiency (IE) of the measured species were used



for nitrate (1.05), organics (1.4) and chlorine (1.3). For the CR data values of 3.39 and 0.88 were used as RIE for ammonium
       and sulfate, respectively. In order to estimate the collection efficiency of the standard vaporizer, we used the chemical-
       dependent-collection-efficiency (CDCE) approach (Middlebrook et al., 2012), with a $NH_4$ detection limit of 0.84, based on
       three times the standard deviation of a 3-hour period in which an HEPA filter was connected before the ACSM inlet. For the
       SKI data the RIE calibration and the estimation of $NH_4$ detection limit were repeated at the beginning and at the end of the

campaign to ensure the accuracy and reliability of the results. This resulted in RIE values of 3.58 for $NH_4$ and 0.90 for $SO_4$
       and a detection limit of 0.53 for $NH_4$ estimated over a 3-hour period.

## 3. Results and discussion

### 3.1 APCI vaporizer temperature evaluation

       The temperature of the APCI vaporizer governs the efficiency of evaporating molecules from the particle phase. The optimal

temperature setting balances incomplete evaporation at lower temperatures and thermal decomposition at higher temperatures
       (Hoffmann et al., 1998; Zhao et al., 2024). To optimize the vaporizer temperature, we performed PAM-OFR experiments with
       α-pinene and $O_3$, to create an aerosol composition over a wide range of particle size, molecular functionality and volatility.
       Once the SOA formation inside the PAM-OFR reached a steady state, the vaporizer temperature was sequentially increased
       from 100 °C to 250 °C, 350 °C, and 450 °C (Fig. 2). The vaporization behavior was evaluated for three compound classes:

$\leqslant C_{10}$, $C_{11}-C_{15}$ and $\geqslant C_{16}$. Different molecular sizes result in distinct vapor pressures and therefore evaporation behavior.
       This is illustrated by the ratio of the total dimer signal intensity ($> C_{10}$) to the total monomer signal intensity ($\leqslant C_{10}$).

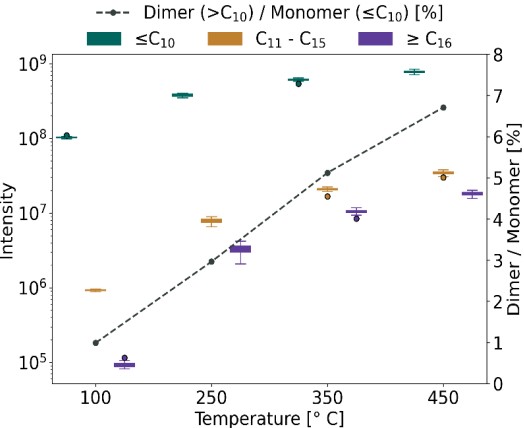

**Figure 2: The effects of varying vaporizer temperatures on the detection of different oxidation products of α-pinene aerosols using APCI-Orbitrap-MS. The measured Intensity are divided into three groups based on their carbon number: $\leqslant C_{10}$ (green), $C_{11}-C_{15}$ (brown) and $\geqslant C_{16}$ (purple). The dashed line indicates the ratio between the detected dimers ($> C_{10}$) and the monomer ($\leqslant C_{10}$).**

       In α-pinene SOA we find a total of 844 compounds, of which 361 compounds were $\leqslant C_{10}$, 228 compounds were $C_{11}-C_{15}$, and
       255 compounds had $\geqslant C_{16}$ with particle mass during these experiments around 300 µg m$^{-3}$. For all compounds, we observed
       an increase in measured intensity when the temperature was increased. Over the measured temperature range, $C_{11}-C_{15}$ and

$\geqslant C_{16}$ compounds increased significantly, by a factor of ~ 40 and 200, respectively. Similarly, the dimer-to-monomer ratio
       increases with temperature. At 100 °C, the fraction of complete aerosol consisting of dimers is less than 1 %, indicating that
       only a small proportion of dimers undergo evaporation. Lower vapor pressure of the dimer compounds requires more energy



for evaporation, which might come at the expense of thermal fragmentation (Glasius and Goldstein, 2016; Zhao et al., 2024). However, we observe an increasing signal of all three compound classes as the temperature is increased, with the dimer-to-

monomer ratio reaching a maximum fraction of 6.7 % at 450 °C. We like to emphasize that the vaporizer temperatures are the temperature set points, and the resulting temperature of the gas stream is certainly below that value. As we are not aiming at recording thermograms, for which precise knowledge on the effective temperatures would be necessary, we were solely evaluating the effectiveness of evaporation and potential signs of thermal decomposition. Finally, we selected an operating temperature of 350 °C for continuous ambient measurements, as the laboratory experiments indicated that at this temperature

monomers and dimers are sufficiently evaporated into gas-phase to be above the instrumental noise level (Fig. S7), while we keep the risk of thermal decomposition low (Hoffmann et al., 1998; Zuth et al., 2018; Zhao et al., 2024) and minimize the heat stress on the ion source. Avoidance of heat stress of the ion source is of particular importance, since we operated the ion source without sheath-, auxiliary-, and sweep gases that cool the ion source. The manufacturer does not recommend to operate the ion source without these gases.

### 3.2 Continuous in-situ performance of the online APCI-Orbitrap-MS


#### 3.2.1 Campus Riedberg (CR) station, an urban background site in Frankfurt, Germany

At CR approximately 2,000 molecular formulas (confidence level 4) were unambiguously identified and subsequently classified into the following compound groups, listed in descending order of intensity: ~1,000 CHO, ~390 CHON, ~480 unclassified formulas (others), ~70 CHOS and ~80 CHNOS compounds (Fig. S8a).

Figure 3a shows the diurnal variation of three monomers $C_6H_{10}O_5$, $C_8H_{12}O_6$ and $C_{10}H_{16}O_3$ (left y-axis) and two dimers $C_{17}H_{26}O_8$ and $C_{19}H_{30}O_5$ (right y-axis) in local time. A distinct diurnal cycle was observed for $C_{10}H_{16}O_3$ (confidence level 4), which we attributed to pinonic acid and its isomers (and further refer to) based on its temporal behavior. We observed high intensities of pinonic acid at night and in the morning, while afternoon levels fell below the instrumental noise threshold, which could be explained by increased temperatures over the day resulting in the partitioning from pinonic acid from particle- into gas-phase.

No clear diurnal pattern was observed for $C_8H_{12}O_6$ (confidence level 2), which we attribute to MBTCA based on its temporal behavior and MS² experiments. MBTCA is a tricarboxylic acid, produced during photooxidation of gas-phase pinonic acid (Szmigielski et al., 2007; Müller et al., 2012; Kostenidou et al., 2018). However, MBTCA exhibited overall fluctuations that were highly correlated with the organic species measured by the ACSM. MBTCA showed slightly higher intensities in the late morning, around noon, with peak intensities on the 08 July at 10:00 h. Coupled with a strong decrease in the afternoon into

the night, before reaching its second peak in the morning of 09 July between 06:00 h and 09:00 h. The BB marker $C_6H_{10}O_5$ (confidence level 2), levoglucosan, behaved similarly to MBTCA, aside from an event on the evening of 09 July. Levoglucosan and its isomers (mannosan and galactosan), are well-established tracer for biomass burning, which is typically formed by the pyrolysis of cellulose (Simoneit et al., 1999). Although literature reports significantly higher levoglucosan concentrations in winter due to domestic heating, substantial average concentrations have also been measured in urban areas during summer.

Studies have reported average concentrations during the summer of 19.1 ng m$^{-3}$ in Ghent (Pashynska et al., 2002), 10–13 ng m$^{-3}$ in suburban areas of Sweden (Mashayekhy Rad et al., 2018) and around 10 ng m$^{-3}$ in Barcelona and Madrid (van Drooge et al., 2014). Elevated levoglucosan signals in summer are often linked to human activities such as barbecuing with charcoal or wood. Which could explain the distinct levoglucosan peak observed on the evening of Sunday, 09 July.

The signal intensities of dimers were overall an order of a magnitude lower than that of the monomers and show no pronounced

diurnal variations. Both dimers, $C_{17}H_{26}O_8$ and $C_{19}H_{30}O_5$, were increasing on 09 July between 06:00 h and 09:00 h. Especially $C_{19}H_{30}O_5$, a dimer product of α-pinene ozonolysis by aldol reaction of cis-pinonic acid and pinonaldehyde or norpinonaldehyde (Lüchtrath et al., 2024), showed a very strong increase during the morning of 09 July.

Figure 3b and 3c compare the OA-related trends of the APCI-Orbitrap-MS with the ACSM organics species, combining the qualitative, molecularly resolved data of the APCI-Orbitrap-MS with the quantitative organics measurements of the ACSM.




The comparison of both instruments over time is shown in Figure 3b as a function of the ACSM organic species (green) and
the extracted ion count (EIC), which is the sum of the extracted signal intensities between $m/z$ 85 to $m/z$ 650 of the APCI-
Orbitrap-MS (black) in negative ionization mode (EIC$_{neg}$). The organic species of the ACSM is mainly comprised of
oxygenated CHO and hydrocarbon CH organic fragments, however it should be noted that, due to the ionization via EI and
the resulting molecular fragmentation in the ACSM, the fragments of organic nitrates, organic sulfates, and organic ammonium

salts are attributed to their respective species of organics, nitrate, sulfate, and ammonium. The $\sum$EIC$_{neg}$ and the concentration
of the ACSM organic [µg m$^{-3}$] were the lowest on 07 July shortly after midnight with ACSM organic concentration of
5.5 µg m$^{-3}$. The highest measurements for both instruments were in the morning of 09 July between 06:00 h and 09:00 h with
22 µg m$^{-3}$.

The observed correlation between the organic species measured by the ACSM and the $\sum$EIC$_{neg}$ exhibited a near linear

290  relationship, as indicated by a Pearson's R value of 0.91 (Fig. 3c). The measured $f_{44}$ of the ACSM organics from CR ranged
mainly between 0.08 and 0.49, with an average of 0.13, indicating less aged aerosol with rather semi-volatile oxygenated OA
(SV-OOA) character (Ng et al., 2010; Ng et al., 2011a). The deviation of APCI-Orbitrap-MS and ACSM organics ranging
from 7.5 to 15 µg m$^{-3}$, can be attributed to less aged OA measured on 08 July between 22:00 h and 24:00 h indicated by a

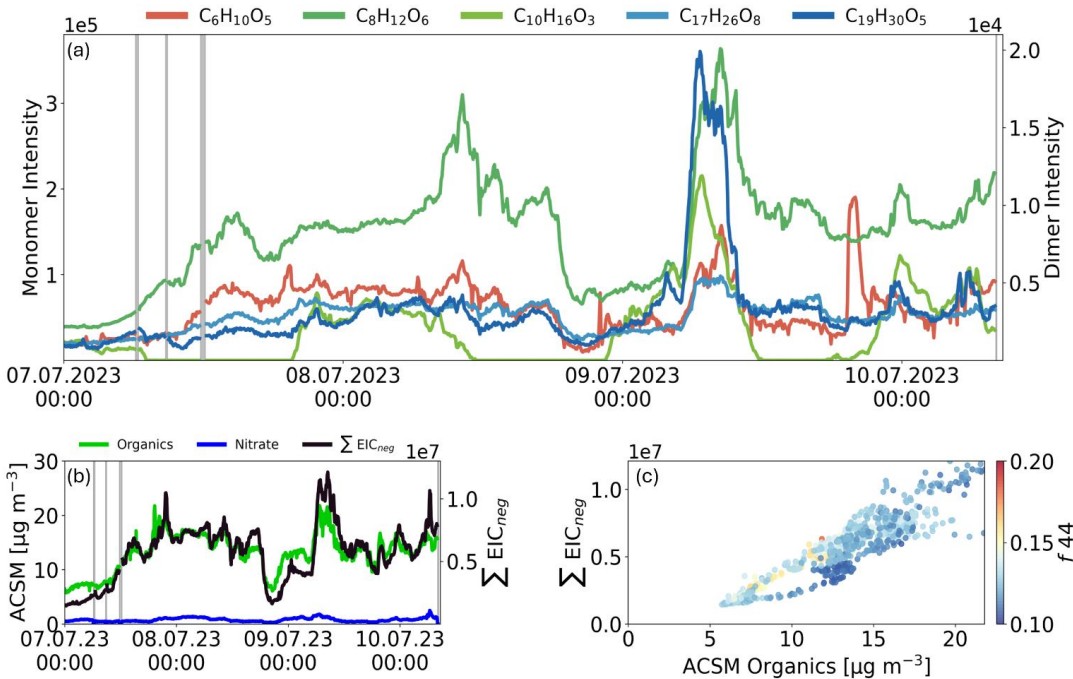

**Figure 3: Ambient air measurements at CR from between the 07 July 2023 00:00 h to 10 July 2023 08:00 h (LT), averaged over 5 min intervals. Grey areas indicate background measurements. a) Timeseries of selected monomers (C$_6$H$_{10}$O$_5$, C$_8$H$_{12}$O$_6$, C$_{10}$H$_{16}$O$_3$) and the dimers (C$_{17}$H$_{26}$O$_8$, C$_{19}$H$_{30}$O$_5$). b) Comparison of ACSM species in [µg m$^{-3}$] for organics (green), nitrate (blue), and $\sum$EIC$_{neg}$ from APCI-Orbitrap-MS (black). c) Pearson correlation of $\sum$EIC$_{neg}$ and ACSM organics species, colored by their respective $f_{44}$ values.**

lower $f_{44}$ value. Typical compounds with a lower $f_{44}$ value are alkanes, alcohols, aldehydes and ketones (Ng et al., 2010), and

295  consequently, have a less acidic character. This makes them less likely to be ionized during APCI-Orbitrap-MS measurements
in negative mode. This finding aligns with the characteristics of the organics measured at CR when considering the triangular
diagram showing the ratio of $f_{44}$ to $f_{43}$ (Fig. S9). The CR measurements fall within the range typically described for ambient
aerosol (dashed lines) (Ng et al., 2010). The $f_{43}$ measured in this study ranges from 0.008 to 0.12 (average 0.08), indicating
that the aerosol exhibits a more semivolatile character.

300



### 3.2.2 Schivenoglia (SKI) station, an agricultural field site in the Italian Po-Valley

The OA at the agricultural field site SKI has a different chemical composition compared to the urban background station CR, as indicated by the continuous measurement in negative ion-mode over a period of seven days (Fig. 4a). Approximately 3,500 molecular formulas were assigned into the following compound groups in descending order of intensity: ~1,100 CHO, ~1,300 305 CHON, ~440 unclassified formulas (others), ~230 CHOS and ~330 CHNOS compounds (Fig. S8b).

At SKI, the aerosol composition is largely influenced by BB-events, with 55 % of all compounds related to BB activities. A distinction can be made here between BB-related compounds that were only detectable during combustion events, such as $C_8H_8O_3$, and compounds that show peak intensities during these events, but remain in particle-phase longer, such as levoglucosan. The consistently elevated intensities of levoglucosan ($C_6H_{10}O_5$, confidence level 2) indicates a constant 310 influence of BB on OA. In addition to BB-derived compounds, the OA in SKI is also significantly influenced by compounds formed through daytime photochemistry and nighttime chemistry. Marker compounds of diurnal photochemistry, such as MBTCA, showed peak concentrations between morning and midday (09:00 h and 14:00 h), when photochemical aging takes place. During the night and early morning hours, the concentration of MBTCA reached its daily minimum, which is due to the low OH radical concentration in the absence of sunlight. The lowest recorded intensities occurred during a rain event on the 315 morning of 04 October. For $C_8H_{13}O_8N$, an organonitrate compound (CHNO) typically formed via nighttime chemistry, clear temporal variability was also observed, with peak intensities at night and in the early morning and lowest values in the afternoon. $C_8H_{13}O_8N$ has been reported in urban environments (Zuth et al., 2018; Guo et al., 2022) and we confirmed by laboratory PAM-OFR experiments its formation by α-pinene ozonolysis in the presence of $NO_x$. During the night the planetary boundary layer (PBL) height is lower leading to more stable conditions and potentially higher concentrations of pollution. At 320 the same time $NO_2$ is not photolyzed but instead reacts with $O_3$ forming $NO_3$, a common nighttime oxidant.

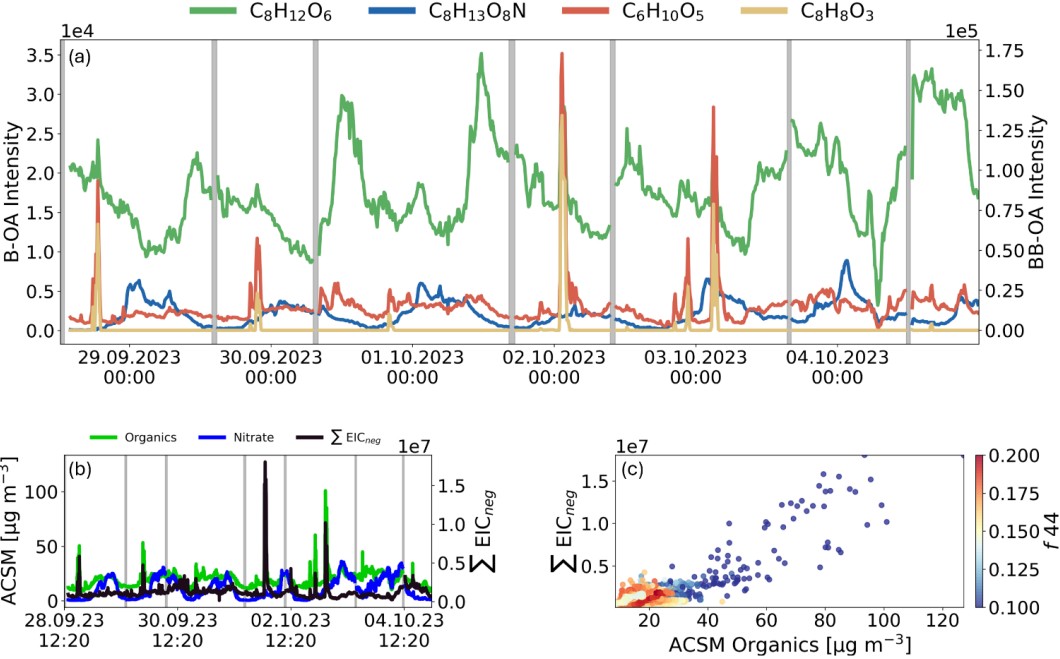

**Figure 4: Ambient air measurements at SKI from 28 Sept 2023 14:00 h to 04 Oct 2023, 24:00 h (LT), averaged over 10 min intervals. Grey areas indicate background measurements. a) Time series of selected biogenic OA (B-OA) tracers ($C_8H_{12}O_6$ in green, $C_8H_{13}O_8N$ in blue) and biomass burning (BB-OA) tracers ($C_6H_{10}O_5$ in red, $C_8H_8O_3$ in orange). b) Comparison of ACSM species in [μg m⁻³] for organics (green), nitrate (blue), and ∑EIC$_{neg}$ from APCI-Orbitrap-MS (black). c) Pearson correlation of ∑EIC$_{neg}$ and ACSM organics species, colored by their respective $f_{44}$ values.**



A comparison of the APCI-Orbitrap-MS with the ACSM (Fig. 4b) reveals that both $\sum EIC_{neg}$ and ACSM organics are substantially influenced by BB-events, which mask the patterns of diurnal variations. The average ACSM organics concentration over the entire measurement period was around 20 µg m⁻³, with peaks reaching up to 127 µg m⁻³ during BB-events. Both $\sum EIC_{neg}$ and ACSM organics are substantially increased during two pronounced BB-events on the 02 October

around 01:30 h (B1) and 03 October at 03:00 h (B2). When comparing these two BB-events, we observe a general decrease of both peaks in B2 compared to B1. Notably, the decrease is more pronounced for $\sum EIC_{neg}$ than the ACSM organics, suggesting that the OA emitted during B2 consists of more hydrocarbon compounds, which are less likely to be ionized with APCI neg mode.

Additionally in SKI submicron nitrate exhibited a pronounced nocturnal pattern, with maximum concentrations occurring

during the night and late morning, reaching up to 35 µg m⁻³ in the morning of 02 October. Nitrate concentrations of less than 1 µg m⁻³ were observed during the daytime hours. This diurnal trend is consistent with the organonitrate measurements from the APCI-Orbitrap-MS, which further supports the observed nitrate dynamics. A similar diurnal behavior was described by Atabakhsh et al. (2023), which reported an average nitrate concentration of 2.16 µg m⁻³ at an agricultural site close to Melpitz (Germany) including peak values between 30 to 40 µg m⁻³ during the winter period (Atabakhsh et al., 2023). The diurnal cycle

of organonitrates can be explained by the nighttime production reaching peak concentrations in the morning hours shortly before vertical mixing and the decrease during the day, primarily driven by temperature-dependent evaporation and reactivity of ammonium nitrate (Stelson and Seinfeld, 1982; Wu et al., 2022). Warmer temperatures during the day accelerate the evaporation of ammonium nitrate ($NH_4NO_3$) due to its high volatility. CHONs, in contrast, can differ in their specific mechanism and may experience reductions due to its chemical reactivity.

Overall, the dependence between ACSM organics and $\sum EIC_{neg}$ (Fig. 4c) exhibited a strong correlation with a Pearson's R of 0.7, which is slightly weaker than at CR. The OA at SKI site can be mainly divided into two groups: Organics with lower $f_{44}$ values ($f_{44} < 0.14$), which account for 15 % of the total ACSM organic species, and consist of fresh aerosol that is mainly released during BB-events, and OA with higher $f_{44}$ values ($f_{44} > 0.14$), accounting for 85 % of the total organic species and associated with more aged and highly oxidized OA (Ng et al., 2010; Ng et al., 2011a). This finding is further supported by the

triangular plot of $f_{44}$ to $f_{43}$ (Fig. S10), which demonstrates the aerosol composition exhibits a relatively constrained $f_{43}$ range between 0.05 and 0.1, while the $f_{44}$ ranges between 0.08 to 0.2 (Ng et al., 2010). This indicates the presence of more photochemically aged compounds in combination with a limited number of fresh emitted compounds.

### 3.3 Importance of high-resolution mass spectrometry for confident molecular formula attribution

The measurements of ambient aerosol, without a chromatographic separation or concentration step, results in highly complex

spectra with low-intensity peaks for both negative and positive mode. Generally, peaks in mass spectra are considered to be poorly resolved when they are less than one full width at half-maximum (FWHM) apart. Insufficient separation can lead to a high margin of uncertainty, potentially causing false molecular attribution or forced integration, which is especially problematic for complex spectra (Müller et al., 2011; Cubison and Jimenez, 2015). Therefore, a high mass resolution enables better baseline separation, which minimizes peak overlap and allows identification of isotopic variants and isobaric species.

This further reduces the uncertainty in molecular formula assignment (Hernández et al., 2012; Nagao et al., 2014; Cai et al., 2021; Yuan et al., 2024).

We visualized the importance of high-resolution mass spectrometry in Figure 5, which displays the number of isobaric interferences per unit mass (top), as well as the measured distances between peaks given in Δ *m/z* (bottom), for negative (left)



and positive (right) ionization mode. Peaks were classified as valid if they had an intensity above 50 and have at least five

consecutive measurement points per peak.

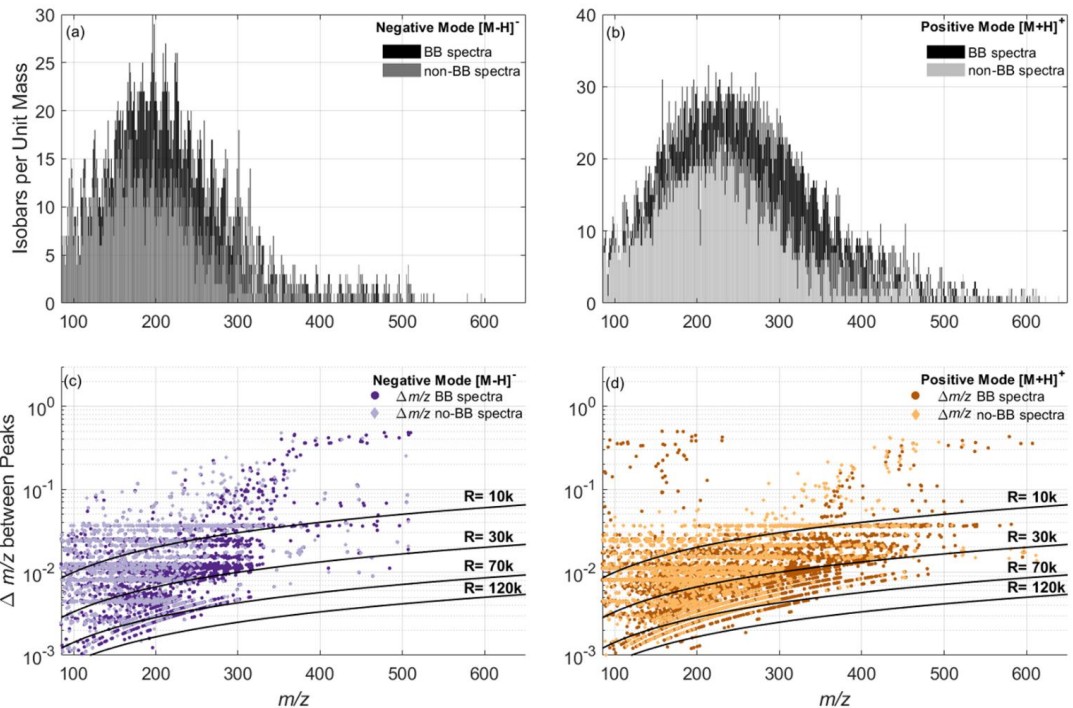

**Figure 5: Ambient BB and non-BB spectra from SKI. Top: Number of isobars per unit mass for negative (5a) and positive (5b) mode. Bottom: Δ*m/z* between isobars for negative (5c) and positive (5d) mode as well as calculated resolutions for R= 10k, 30k, 70k and 120k.**

The spectra shown were recorded during a BB event and for non-BB conditions in both polarities (Fig. 5a and 5b). During the BB event, the highest number of isobaric peaks per unit mass were observed with up to 30 peaks at *m/z* 196 in negative mode, and up to 33 peaks at *m/z* 214 in positive mode. During non-BB conditions, the number of isobaric peaks per unit mass was

lower. In negative mode a maximum of 18 peaks per unit mass were observed at *m/z* 166 and 198, while in positive mode up to 24 peaks per unit mass were detected at *m/z* 214, 221 and 226. Hence more complex spectra during the BB events have a larger number of isobaric peaks across the whole spectrum.

Figure 5c and 5d show the average Δ *m/z* between measured peaks on the nominal mass and simulated resolutions for R= 10k, 30k, 70k, and 120k. These plots visualize the resolution required to distinguish peaks by at least one FWHM in ambient

samples, which is critical for confident molecular formula attribution. Consistent with the isobars per unit mass trend, the smallest Δ *m/z* values are observed between *m/z* 85 and 300, where peak density is highest. For instance, a minimum peak distance of $0.89 \times 10^{-3}$ was detected at *m/z* 99 in negative ionization mode. As the number of peaks decreases above *m/z* 300, the average distance between peaks increases. The smallest Δ *m/z* above *m/z* 300 is 0.0046. A similar trend is observed in positive mode, with the narrowest peak separation of $0.88 \times 10^{-3}$ at *m/z* 89. Overall, the Δ *m/z* between peaks per unit mass

spans a wide range. While some peaks could be resolved with a resolution below 10k, 70k is required for achieving sufficient FWHM peak separation across the full spectrum, especially up to *m/z* 250 in negative mode and *m/z* 350 in positive mode. In our ambient measurements, a resolution of R= 120k appears necessary in order to perform unambiguous attribution of molecular formulas, as only 15 peaks fall below the simulated R= 120k resolution line. However, this conclusion is constrained





by our instrument, which operated at R = 120k. Meaning we cannot rule out that overlapping peaks are present that would

require R > 120k to be separated.

We demonstrate the differences of resolution using levoglucosan as an example, as a relevant compound especially in highly BB-influenced areas. Figure 6 shows spectra of levoglucosan recorded during BB events in negative (top) and positive (bottom) ionization mode. In negative ionization mode, the highest-intensity peaks are observed at lower $m/z$ values up to ~ 150, with the peak intensity gradually decreasing towards $m/z$ 200. This range includes a majority of monoterpene and sesquiterpene

oxidation products, in addition to numerous BB markers. Furthermore, some dimers are detected, primarily within the $m/z$ range of 250 to 500. In positive ionization mode, while low $m/z$ values similarly exhibit the highest intensities, significant higher intensity peaks are detected in the $m/z$ range from 400 to 600.

Around the peak of levoglucosan, at $m/z$ 161.0455 in negative mode (Fig. 6b) and $m/z$ 163.0502 in positive mode (Fig. 6c), we detected 7 and 12 neighboring peaks with a narrow mass range of one unit mass, in negative and positive ionization mode

respectively. In negative ion-mode, eight peaks were distinctly baseline-separated. In addition to the high resolution we find a high mass accuracy with mass differences between the measured and theoretical $m/z$ values ranging from -0.02 to -0.19 ppm. Due to the high resolution and accurate mass of the Orbitrap-MS, seven of these peaks could be identified with an assigned molecular formula (Tab. S1). In positive ion-mode, a total of 13 peaks were baseline-separated with mass differences ranging from 0.16 and 0.65 ppm (Tab. S2).

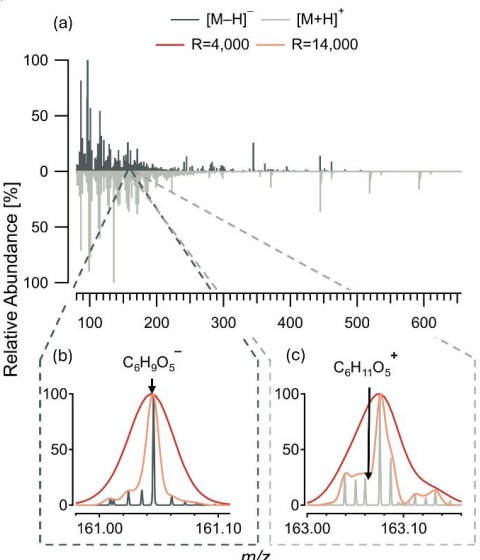

**Figure 6: Relative abundance [%] for a) full MS spectra between *m/z* 85 to 650 for negative (dark grey, 02.10.2023 22:00–03.10.2023 04:00 LT) and positive (light grey, 09.10.2023 22:00–10.10.2023 04:00 LT) ionization mode from ambient measurements in SKI. b+c) Spectra section including $C_6H_{10}O_5$ in negative and positive mode respectively, with the peak intensity and shape for Orbitrap (R= 120,000) and simulated mass resolutions of R= 4,000 (red) and R= 14,000 (orange) representing average ToF-MS instruments.**

Notably, in positive ionization mode, the levoglucosan ion ($m/z$ 163.0602) did not exhibit the highest intensity within the isolation window, but instead the neighboring peak corresponding to $C_{10}H_{11}O_2^+$ ($m/z$ 163.0755) was more prominent. The separation between these peaks is minimal, with a mass difference of 0.0153 Da, a span that instruments with lower resolution, such as R = 4k to R = 14k (Riva et al., 2019), encounter significant challenges in resolving. In negative ionization mode, a lower resolution instrument could correctly identify $C_6H_9O_5^-$ as it represents the highest-intensity compound. However, the

lower resolution compared to the Orbitrap-MS could potentially result in misinterpretation or omission of $C_6H_{11}O_5^+$, due to its





location within the shoulders of the more prominent $C_{10}H_{11}O_2^+$ peak. Even at R = 14k, while the $C_6H_{11}O_5^+$ peak would be partially distinguishable from the shoulder, it would still overlap with other peaks of similar intensities of $C_8H_7O_2N_2^+$ (*m/z* 163.0502) and $C_9H_7O_3^+$ (*m/z* 163.0389), making accurate identification challenging.

This highlights the need for high resolution, to unambiguously assign chemical formulas, which can only be done if peaks are
sufficiently separated and do not lie within the shoulder of a neighboring peak, as shown for levoglucosan in positive mode. Further, the high mass resolution also allows the determination of higher abundance isotopes like $^{34}$S and $^{14}$C, however, lower abundant isotopes like $^{18}$O and $^{15}$N are not sufficiently detected, which would be needed in order to detect the isotopic fine structure of nitrogen and oxygen-containing compounds (Nagao et al., 2014; Dittwald et al., 2015).

### 3.4 Instrumental sensitivity of the APCI-Orbitrap-MS


In addition to the importance of high mass resolution, we also want to highlight the versatility of the APCI-Orbitrap-MS for its ability to detect compounds over a wide chemical range. The sensitivity of the instrument for a specific compound is based on the ionization efficiency, the ion transmission efficiency, and the detection performance. To test the sensitivity of the APCI-Orbitrap-MS, we investigated different biogenic (B-OA), anthropogenic (A-OA) and biomass burning organic aerosol (BB-OA) compounds in both negative [M−H]⁻ and positive ionization mode [M+H]⁺ (Fig. 7).

We normalized the signal intensity of the MS by the measured particle mass [µg m⁻³] recorded by an SMPS. While this approach theoretically allows for the calculation of calibration factors to gain quantitative insights for each compound, its application to ambient measurements is limited. This is primarily due to the limitation of reference standard representative of

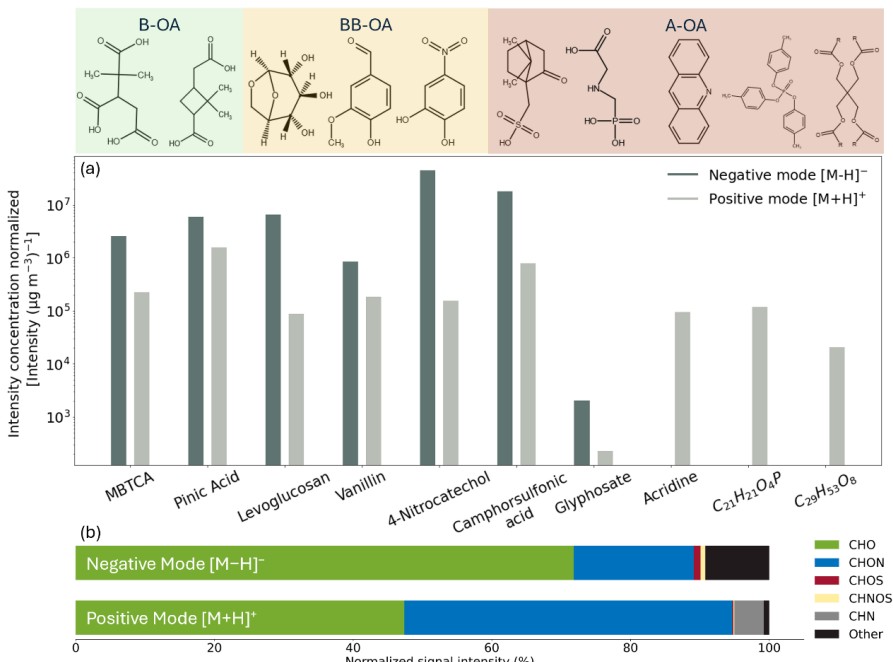

**Figure 7: APCI-Orbitrap-MS ionization experiments of reference standards for biogenic (B-OA, green), biomass burning (BB-OA, brown), and anthropogenic (A-OA, red) organic aerosols. a): Sensitivity of the APCI-Orbitrap-MS according to the ionization efficiencies in both negative ionization mode [M-H]⁻ (dark grey) and positive ionization mode [M+H]⁺ (light grey) for B-OA (MBTCA and pinic acid), BB-OA (levoglucosan, vanillin, and 4-nitrocatechol) as well as anthropogenic (camphor sulfonic acid, glyphosate, acridine, and two compounds included in Mobil Jet™ Oil II ($C_{21}H_{21}O_4P$ and $C_{29}H_{53}O_8$)). The measured intensities are normalized to their respective particle mass [Intensity (µg m⁻³)⁻¹] measured by an SMPS. b) Distribution of the normalized signal intensity form SKI measurements for each compound group in both negative [M-H]⁻ (28.09.–05.10.2023) and positive ionization mode [M+H]⁺ (05.10.–12.10.2023).**



ambient OA, as well as differences between the experimental laboratory setup and field conditions, which are likely caused by
matrix effects and possible losses in the inlet system.

All experimentally investigated B-OA and BB-OA compounds, along with camphorsulfonic acid and glyphosate (A-OA), showed higher sensitivity in APCI negative ion-mode compared to the positive ion-mode. 4-nitrocatechol had the highest ionization efficiency in negative ion-mode followed by camphorsulfonic acid, pinic-acid, levoglucosan and MBTCA.
Explained by their elevated gas-phase acidity, which is associated with the presence of electrophilic functional groups, such as carboxylic acids among others (Carroll et al., 1975; Sharon and Bartmess; Derpmann et al., 2014). Although levoglucosan and vanillin lack carboxylic acid groups, their alcohol groups provide a certain degree of acidity given by their electronegativity, allowing them to ionize more efficiently in negative than in the positive ion-mode, as described in the literature (Gambaro et al., 2008; Staš et al., 2017; Carregosa et al., 2023). Acridine and two components of Mobil Jet™ Oil II
($C_{21}H_{21}O_4P$ and $C_{29}H_{53}O_8$), on the other hand, were only ionizable in positive ion-mode, highlighting the importance of measuring in both polarities.

When examining the differences in the distribution of compound classes in ambient air samples measured in SKI (Fig. 7b), it highlights the importance of using both positive and negative ionization modes to obtain a more comprehensive understanding of the chemical composition of ambient OA (Tab. S3). In negative ionization mode (recorded between 28 September and 05
October 2023) CHO compounds dominated with a total of 1,112 attributed formulas and an overall $\sum$intensity of $9.22\times10^9$. In positive ion-mode (recorded between 05 and 12 Oktober 2023) we attributed a very similar number of CHO formulas (1,115), but with only half of the $\sum$intensity, specifically $4.84\times10^9$. For the CHON group, 1,287 individual compounds were detected in negative ion-mode, with a total intensity of $2.2\times10^9$, while in positive ion-mode, where 1,269 substances were identified, with more than double in total intensity yielding $4.82\times10^9$ CHOS and CHNOS compounds were mostly detected in negative
ionization mode, while CHN compounds were only detected in positive ion-mode, with 155 formulas and a total $\sum$intensity of $4.34\times10^8$.

### 3.5 Online laboratory and ambient fragmentation experiments

To further improve data quality and confidence of chemical formula assignment, we demonstrated the feasibility of the APCI-Orbitrap-MS to perform online fragmentation experiments for ambient and laboratory aerosol. This step improves the
confidence level of the tested compounds from level 4, unequivocal molecular formula, to level 2, probable structure, according

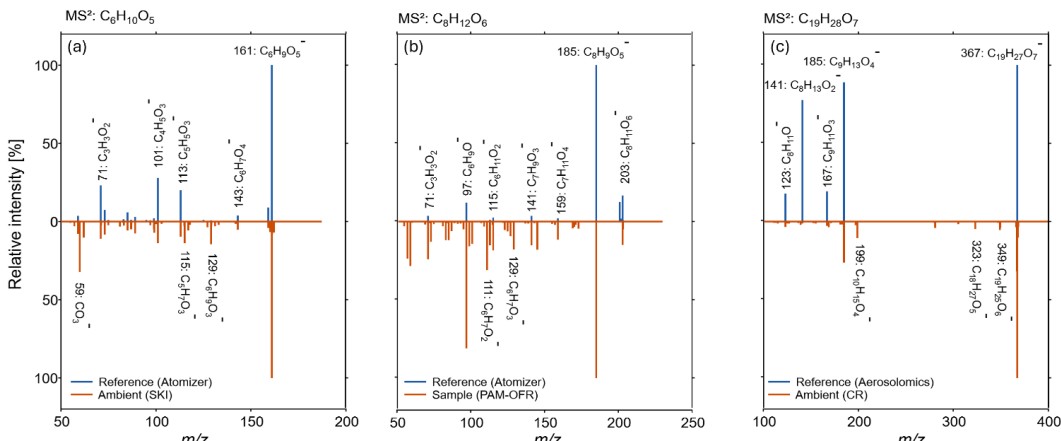

**Figure 8: Mirror spectra for online MS² experiments of reference measurements or spectral libraries (top spectra respectively) and complex ambient or laboratory aerosol mixtures (bottom spectra respectively) for a) $C_6H_{10}O_5$, b) $C_8H_{12}O_6$ and c) $C_{19}H_{28}O_7$.**



to Schymanski et al. (2014). A notable advantage of the APCI-Orbitrap-MS setup over other CI methods using reagent ions is its ability to perform $MS^2$ experiments. Traditional CI methods that charge the analyte compounds by clustering with reagent ions often face the challenge of the loss of the reagent ions in the fragmentation experiments, while no fragmentation of the target molecules is achieved.

The main challenges encountered during online $MS^2$ experiments using the APCI-Orbitrap-MS is the lack of an upstream separation technique, resulting in the fragmentation of every compound within the isolation window. This increases the risk of isobaric compounds with similar fragmentation patterns interfering with the analysis. Despite this, accurate identification is still possible by comparing the spectra to those recorded with standard compounds or to entries in spectral libraries.

For the evaluation of ambient $MS^2$ experiments, we compared the fragmentation pattern of three different reference compounds
(top) to a complex aerosol mixture from ambient or laboratory measurements (bottom) in Figure 8. This comparison was conducted using the reference standards introduced for $MS^2$ experiments with levoglucosan (Fig. 8a) and MBTCA (Fig. 8b). The fragmentation spectra for hydroxypinonyl ester of cis-pinic acid (Fig. 8c) was obtained from the Aerosolomics database (Thoma et al., 2022).

In the $MS^2$ experiments of levoglucosan (confidence level 2), $m/z$ 161 ($C_6H_9O_5^-$, $[M - H]^-$) was identified as the molecular
ion. The most abundant fragment ion signals were at $m/z$ 101 ($C_4H_5O_3^-$, $[M - H - CH_3OH - CO]^-$), $m/z$ 71 ($C_3H_3O_2^-$, $[M - H - C_2H_4 - CO_2 - H_2O]^-$) and $m/z$ 113 ($C_5H_5O_3^-$, $[M - H - CH_4O_2]^-$). This fragmentation pattern aligns with $MS^2$ experiments conducted during field measurements in SKI, where we detected the same ion signals in similar relative ratios. Given the agreement of the fragmentation patterns between the SKI sample and the standard compound measurements, it can be deduced that the fragmentation was mainly from levoglucosan. Additional fragments with high intensities recorded during field
measurements were $m/z$ 59 ($CO_3^-$), $m/z$ 129 ($C_6H_9O_3^-$, $[M - H - O_2]^-$) and $m/z$ 115 ($C_5H_7O_3^-$). The latter could possibly be attributed to the molecular ion $C_7H_{13}O_4^-$, which is with a difference of 0.03 amu located within the isolation window of $m/z$ 1 around $C_6H_9O_5^-$ and could come from the loss of $H_2O$ and $C_2H_4$.

During $MS^2$ experiments of the tricarboxylic acid MBTCA (confidence level 2) $m/z$ 203 ($C_8H_{11}O_6^-$, $[M - H]^-$) was identified as the molecular ion. The main fragments, with decreasing relative abundance were $m/z$ 185 ($C_8H_9O_5^-$, $[M - H_2O]^-$), $m/z$ 97
($C_6H_9O^-$, $[M - H_2O - 2CO_2]^-$), $m/z$ 141 ($C_7H_9O_3^-$, $[M - H_2O - CO_2]^-$) and $m/z$ 115 ($C_6H_{11}O_2^-$, $[M - 2CO_2]^-$), which agrees which in literature reported $MS^2$ fragments (Müller et al., 2012; Zuth et al., 2018; Thoma et al., 2022; Szmigielski et al., 2007). Hereby the loss of $H_2O$ and $CO_2$ is characteristic for the loss of carboxylic acid functional groups. These characteristic fragments can also be found in the fragmentation experiment of MBTCA in the complex SOA oxidation mix created with the PAM-OFR, which showed the same order of relative abundance of fragment ions. However, more spectra were detected, which
could be attributed to compounds within the isolation window of $m/z$ 1 like $C_9H_{15}O_5^-$ and $C_{12}H_{11}O_2^-$, which also will undergo fragmentation.

The hydroxypinonyl ester of cis-pinic acid ($m/z$ 367, $C_{19}H_{27}O_7^-$, confidence level 2) was already in detail described and reported from chamber experiments by Hoffmann et al. (1998) and Kahnt et al. (2018), however to our knowledge, our observation is the first online detection (Fig. S13) of these dimers (Kahnt et al., 2018). Due to the lack of reference standards
a laboratory generated $MS^2$ spectra was not possible, however the fragmentation pattern is included in the Aerosolomics spectral library. Here the main reported fragments were $m/z$ 185 ($C_9H_{13}O_4^-$, $[M - C_{10}H_{14}O_3]^-$) which is cis-pinic acid, $m/z$ 167 ($C_9H_{11}O_3^-$, $[M - C_{10}H_{14}O_3 - H_2O]^-$), $m/z$ 141 ($C_8H_{13}O_2^-$, $[M - C_{10}H_{14}O_3 - CO_2]^-$) and $m/z$ 123 ($C_8H_{11}O^-$, $[M - C_{10}H_{14}O_3 - H_2O - CO_2]^-$). We also detected these fragments during $MS^2$ experiments of ambient OA with $m/z$ 185 being the most dominant fragment. However, in the ambient air experiment, the second most abundant fragment is $m/z$ 199 ($C_{10}H_{15}O_4^-$, $[M - C_9H_{12}O_7]^-$)
which results from the ester cleavage into cis-pinic acid and $C_{10}H_{15}O_4^-$. Aside from this, the main fragments in ambient air were $m/z$ 349 ($C_{19}H_{25}O_6^-$, $[M - H - H_2O]^-$) and $m/z$ 323 ($C_{18}H_{27}O_5^-$, $[M - H - CO_2]^-$).

Although $MS^2$ experiments offer a more specific evaluation of single ions, the lack of upstream separation and currently no automatized trigger for $MS^2$ spectra acquisition makes these experiments elaborate. Automized $MS^2$ after ion mobility



separation might therefore be a way to record cleaner MS² spectra in the field to further push online measurements toward

unambiguous identification. However, this effort is likely only justified for extremely relevant compounds, e.g. pesticides, illicit drugs or explosives.

## 4.   Conclusion and outlook

This study successfully demonstrated the in-situ application of APCI-Orbitrap-MS for real-time, high-resolution and accurate mass measurements of ambient OA composition, addressing key limitations of commonly used online techniques such as fragmentation, low time resolution and low mass resolution. Additionally, it overcomes some challenges associated with offline filter collection, including sampling, storage and workup artifacts.

As proof of principle, the APCI-Orbitrap-MS was deployed during summer 2023 inside an MLC at an urban background

station (CR) and an agricultural field site (SKI), where it showed strong agreement with ACSM organic measurements. Using the APCI-Orbitrap-MS, we observed distinct differences in OA composition between CR and SKI. At CR, no clear diurnal trend was found for total OA measured by the APCI-Orbitrap-MS or Org species concentrations from the ACSM, which ranged between 5.5 and 22 µg m⁻³. However, the molecularly resolved analysis provided by the APCI-Orbitrap-MS revealed individual compounds with distinct diurnal patterns, such as pinonic acid. Generally, the OA composition at CR was dominated

by CHO compounds, showing strong agreement with ACSM organics measurements, with a Pearson correlation coefficient of 0.91. In contrast, SKI exhibited more pronounced diurnal patterns for organic compounds and a higher proportion of CHON compounds. On average, a good agreement between APCI-Orbitrap-MS and ACSM organic was observed, with a Pearson correlation coefficient of 0.7. ACSM organic concentrations were on average higher than at CR, with an average concentration of 20 µg m⁻³ and peak concentrations of 127 µg m⁻³ during BB-events. These events released a large number of compounds,

such as vanillin, which could only be detected for short periods. The frequent, short-lived nature of these BB-events highlighted the importance of high temporal resolution measurement.

The field measurements demonstrated the complexity of OA aerosols and the need for high mass resolution analysis to effectively separate peaks and confidently assign molecular formulae. A series of laboratory experiments were carried out to further understand ambient data. The importance of high resolution was demonstrated for complete mass range in negative

and positive ionization mode and by the example of levoglucosan, which would be difficult to detect with lower resolution instruments, because during BB-events levoglucosan appeared in the shoulder of a neighboring peak in positive mode. The high resolution of the Orbitrap (R = 120k at *m/z* 200) was able to baseline separate these peaks, allowing for unambiguous detection of levoglucosan. The resolving power was sufficient for resolving the isotopic fine structure of $^{13}C$ and $^{34}S$ isotopes, but not for $^{15}N$ and $^{18}O$-containing isotopes, which would further increase confidence levels, with which molecular formulas

are assigned. Future studies could address this issue by deploying instruments with even higher resolution for field measurements.

Despite this, further improvements to data quality were achieved through conducting additional laboratory experiments. We assessed the instrument's sensitivity across a diverse range of biogenic, anthropogenic, and BB-related compounds. All tested biogenic and BB-related compounds, along with the anthropogenic compound's camphor sulfonic acid and the pesticide

glyphosate, exhibited higher sensitivities in negative ionization mode. However, the experiment also revealed that certain anthropogenic compounds, such as those found in Mobil Jet™ Oil II and acridine, were only ionizable in positive ionization mode. This finding highlights the necessity of utilizing both ionization modes to achieve a more comprehensive understanding of OA composition. To further validate molecular assignments and overcome challenges related to isobaric interferences, MS²



experiments were successfully conducted on the field site and later compared to reference standards, successfully improving
confidence levels from level 4 to level 2.

In conclusion, this study highlights the feasibility and versatility of APCI-Orbitrap-MS for field deployments, providing molecular-level insights into atmospheric processes. By expanding its use in field studies, this technology could extend capabilities of widely used aerosol measurement devices, offering enhanced data quality and a deeper understanding of the qualitative aerosol composition. Additionally, we demonstrated the potential of the APCI-Orbitrap-MS for qualitative
measurements by using reference standards. Further studies will be required to be able to put this approach into practice, due to remaining challenges of limited availability of reference standards and the transferability of laboratory-based calibrations to ambient setups.

**See also the supplement to this manuscript.**


**Author contribution**

JD wrote the paper, with ALV advising on the writing process. JD, LD and ALV conceptualized the study. JD, LD, MS, and ALV contributed to the laboratory and field measurements. Orbitrap data evaluation was performed by JD, and LD processed the ACSM data. ALV led the project administration. All authors provided feedback on the paper and contributed to the
scientific discussions.

**Competing interests**

The authors declare that they have no conflict of interest.

**Acknowledgements**

We want to thank Giorgio Siliprandi, Lorenzo Mari and Enrico Bicelli from ARPA Lombardia for their support during the SKI campaign. We also want to thank Florian Ungeheuer for synthesizing MBTCA and Pinic acid.

**Financial support**

This research has been supported by the Deutsche Forschungsgemeinschaft (DFG; German Research Foundation) (grant no.
555 410009325).

This open-access publication was funded by the Goethe University Frankfurt.

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
