# Peer review of "Real-time organic aerosol characterization via Orbitrap mass spectrometry in urban and agricultural environments"

_EGUsphere, 2025_

## Author Comment (AC1)

**Reply to anonymous Referee #1**

We thank referee #1 for the constructive comments, which improved the manuscript. Original comments are written in black, our replies in blue as well as comprehensible excerpts from the text highlighting the tracked changes.

1.Lines 86-88: The authors state in their summary of existing online measurement techniques for particulate organic components that VIA experiences fragmentation at 250°C, emphasizing that the APCI-Orbitrap-MS mitigates this fragmentation. Figure 2 depicts the effect of temperature on signal intensity for different compounds. However, this figure alone does not demonstrate that fragmentation in the APCI-Orbitrap-MS is less severe than in VIA. Thermal decomposition/fragmentation remains a key challenge for the online measurement of organic components in particulates. The argument presented here would be substantially strengthened if the authors could include a direct comparison of thermal decomposition between the two instruments or provide a comparative analysis against relevant published data in the literature.

Response:
We acknowledge that Figure 2 does not provide sufficient information about thermal fragmentation. The primary focus of this figure was to determine optimal vaporizer temperatures for efficiently evaporating molecules from the particle phase, using a complex SOA matrix. While we did monitor specific tracers, including those highlighted by Zhao et al. (2023), and did not observe any thermal fragmentation during our experiments, we recognize that the potential for fragmentation can vary based on the stability of each individual compound against thermal decomposition. Certainly, compounds like peroxides are prone to thermal decomposition also with the presented method.
To address the reviewer's concern, we will revise the X-axis description of Figure 2 to clarify that the vaporizer temperatures represent set points, and the actual temperature of the gas stream is expected to be lower. Additionally, we will include additional explanations that highlight this issue of thermal stability and the potential for fragmentation.

recording thermograms, for which precise knowledge on the effective temperatures would be necessary, we were solely
245    evaluating the effectiveness of evaporation and potential signs of thermal decomposition. Additionally, even though not observed during this experiment (Fig. S7) the thermal decomposition of compounds cannot be ruled out, as it is dependent on the structural stability of each individual compound. Finally, we selected an operating temperature of 350 °C for continuous ambient measurements, as the laboratory experiments indicated that at this temperature monomers and dimers are sufficiently

2.Figure 7: I have reservations about the normalization of units applied by the authors in this figure. This normalization obscures true concentration levels and makes it impossible to assess the original mass spectral intensity information. Consequently, the processed data significantly hinders its utility for meaningful interpretation and fails to support critical information assessment.

Response:
We appreciate your feedback regarding the normalization of units applied in Figure 7. We understand your concern that this normalization may obscure true concentration levels and hinder meaningful interpretation of the original mass spectral intensity data. To clarify, the

aerosol used in our study was generated in the laboratory, which resulted in considerable variation in aerosol mass. To facilitate a meaningful comparison of instrument sensitivity to specific compounds, we normalized the data to account for the differences in compound-specific sensitivity per aerosol mass.

While we believe that this normalization approach is essential for comparing the response of the different compounds per aerosol mass, we also acknowledge the importance of providing complete transparency in our data presentation. Therefore, we will include a table in the SI to allow for a comprehensive understanding of the original mass spectral intensity levels.

Table S3: Instrumental sensitivity experiments with different reference standards and their respective concentration in standard solution [ug L−1], average signal background intensity, average signal intensity and produced mass concentration [µg m−3] used for normalization of Fig. 7.

| Compound | Polarity | Concentration in standard solution [ug L$^{-1}$] | Average signal background intensity | Average signal intensity | Produced particle mass concentration [µg m$^{-3}$] |
|---|---|---|---|---|---|
| MBTCA | [M−H]$^-$ | 5.11 | $1.55 \times 10^5$ | $9.11 \times 10^7$ | 35.43 |
| MBTCA | [M+H]$^+$ | 5.11 | $1.69 \times 10^4$ | $8.03 \times 10^6$ | 35.43 |
| Pinic Acid | [M−H]$^-$ | 2.4 | $1.22 \times 10^6$ | $1.17 \times 10^7$ | 1.77 |
| Pinic Acid | [M+H]$^+$ | 2.4 | $1.72 \times 10^5$ | $2.95 \times 10^6$ | 1.77 |
| Levoglucosan | [M−H]$^-$ | 1.2 | $1.85 \times 10^5$ | $3.02 \times 10^6$ | 0.44 |
| Levoglucosan | [M+H]$^+$ | 1.2 | $1.74 \times 10^3$ | $3.91 \times 10^4$ | 0.42 |
| Vanillin | [M−H]$^-$ | 116 | $7.99 \times 10^5$ | $2.69 \times 10^6$ | 2.23 |
| Vanillin | [M+H]$^+$ | 116 | $3.56 \times 10^4$ | $4.48 \times 10^5$ | 2.22 |
| Nitrocatechol | [M−H]$^-$ | 3.2 | $2.18 \times 10^6$ | $2.87 \times 10^7$ | 0.60 |
| Nitrocatechol | [M+H]$^+$ | 3.2 | $1.12 \times 10^3$ | $1.12 \times 10^5$ | 0.72 |
| Camphorsulfonic acid | [M−H]$^-$ | 3.3 | 0 | $7.70 \times 10^7$ | 4.28 |
| Camphorsulfonic acid | [M+H]$^+$ | 3.3 | $9.19 \times 10^3$ | $3.37 \times 10^6$ | 4.26 |
| Glyphosate | [M−H]$^-$ | 3.5 | 0 | $1.06 \times 10^5$ | 52.37 |
| Glyphosate | [M+H]$^+$ | 3.5 | 0 | $1.20 \times 10^4$ | 52.37 |
| Acridin | [M+H]$^+$ | 1.6 | 0 | $8.06 \times 10^3$ | 0.09 |
| C$_{21}$H$_{23}$O$_4$P | [M+H]$^+$ | 2.1 | 0 | $2.44 \times 10^6$ | 20.86 |
| C$_{29}$H$_{53}$O$_8$ | [M+H]$^+$ | 2.1 | 0 | $4.28 \times 10^5$ | 20.86 |

3.Line422-423: "4-nitrocatechol had the highest ionization efficiency". This does not necessarily reflect ionization efficiency, but could also be affected by ion transmission efficiency.

Response:
We acknowledge that the phrasing "4-nitrocatechol had the highest ionization efficiency" should more accurately be the instrument sensitivity rather than ionization efficiency alone.

435  matrix effects and possible losses in the inlet system.

All experimentally investigated B-OA and BB-OA compounds, along with camphorsulfonic acid and glyphosate (A-OA), showed higher sensitivity in APCI negative ion-mode compared to the positive ion-mode. The highest instrumental sensitivity showed 4-nitrocatechol had the highest ionization efficiency in negative ion-mode followed by camphorsulfonic acid, pinic-acid, levoglucosan and MBTCA. Explained by their elevated gas-phase acidity, which is associated with the presence of

440  electrophilic functional groups, such as carboxylic acids among others (Carroll et al., 1975; Sharon and Bartmess; Derpmann

4.Figure S7: The concentration of organic molecules is a critical limiting factor for online structural identification of compounds. The figure indicates that the maximum intensity of organic molecule signals reaches 1.8E7. This represents an exceptionally high signal level for online measurements. What causes such elevated signal intensities in organic molecules, and whether the authors implement optimizations to enhance instrument sensitivity?

Response:

We agree with Reviewer #1 that during the experiments elevated signal intensities were reached. When again looking into the raw data the TIC reached 1.15E9, which is definitely a high signal, but is still within the linear range of the instrument and not at risk for oversaturation of the detector.

The experiment Figure S7 is based on SOA generated from the ozonolysis of α-pinene, achieving an average particle mass concentration of 306.6 µg m$^{-3}$ with a laboratory setup. While the SOA production in the laboratory often is only a simulation of ambient processes, accurate levels of precursors in an oxidative system are challenging to achieve and maintain. However, during the field campaigns the concentration of organics measured by the ACSM reached up to 127 µg m$^{-3}$. This indicates that our experimental conditions adequately cover the expected range of aerosol mass concentrations.

We recognize the importance of providing more detailed information regarding the optimization of instrument sensitivity. To address this, we have included several key strategies in the revised section '3.4 Instrumental Sensitivity of the APCI-Orbitrap-MS'.

**3.4 Instrumental sensitivity of the APCI-Orbitrap-MS**

420    In addition to the importance of high mass resolution, we also want to highlight the versatility of the APCI-Orbitrap-MS for its ability to detect compounds over a wide chemical range. The sensitivity of the instrument for a specific compound is based on the ionization efficiency, the ion transmission efficiency, and the detection performance. To enhance the instrument sensitivity, particularly for a targeted approach, conducting online measurements in selected ion monitoring (SIM) mode can significantly improve sensitivity by focusing on one or a few selected $m/z$. Additional sensitivity improvements can be achieved

425    by optimizing the radio frequency (RF) amplitude applied to the S-lense and the automatic gain control (AGC). Adjusting the RF amplitude enhances ion transmission in specific mass ranges, while optimizing the AGC target value controls the number of ions accumulated in the C-trap for subsequent introduction into the mass analyzer. To test the sensitivity of the APCI-

---

## Author Comment (AC2)

**Reply to anonymous Referee #2**

We would like to thank Referee #2 for the constructive comments, which have significantly enhanced the quality of our manuscript. The original comments are presented in black, while our responses are provided in blue, accompanied by excerpts from the text that illustrate the tracked changes. We have thoroughly revised the entire document, carefully considering all feedback in our efforts to improve the manuscript. While we have included screenshots to demonstrate respective changes based on the reviewer's comments, please note that subsequent modifications may not be documented with screenshots.

Overall comments

- I would change the structure of the results section. The field data is interesting but it would be much better if this followed on from the technical descriptions of the importance of high resolution, sensitivity and the use of fragmentation. For instance, there is a discussion in the fragmentation section about the assignment of MBTCA that I would have appreciated to understand before looking at the field observed data. Then I could take into account the discrepancies between the lab generated spectrum and the reference spectrum. The importance of having high resolution and the number of compounds that can be resolved at each nominal mass is one of the most important features of the new method and I think it would be more appropriate to have this as the first part of the results section.

    Response:

    We appreciate your feedback regarding the structure of the results section in our manuscript. We appreciate your suggestion to prioritize the technical descriptions related to high resolution, sensitivity, and fragmentation before presenting the field data. We initially structured the results and discussion in the way you suggested, however during internal discussions we have decided to change it to its current structure, as the primary objective of our study is to demonstrate the successful field deployment of the Orbitrap technology, which we believe is a crucial message of our findings. By presenting the time series of individual molecules early in the manuscript, we aimed to highlight the practical applications and significance of our method in real-world scenarios. Additionally, many subsequent analyses and results refer back to this field data (Fig. 5, Fig. 6, Fig. 7b, Fig. 8). That being said, we understand the value in providing a clearer context to the importance of high resolution and the fragmentation of MBTCA earlier in the manuscript, before showing and discussing the field observations, however we think as stated above would this only lead to more open questions as to where all the data for subsequent analysis are coming from.

- There is very little presented on the limitations of the technique. I would suggest adding a paragraph outlining the issues and how/if they could be overcome in future studies. The isobaric interferences is mentioned but there are other limitations around calibration and unassigned peaks.

Response:

Thank you for your valuable feedback regarding the limitations of the technique discussed in our manuscript. We appreciate your suggestion to elaborate on these challenges, which in our opinion significantly improves the overall manuscript and helps the reader so put it better into context.

In response, we have added a paragraph to the conclusion that outlines several key challenges associated with the APCI-Orbitrap-MS method. Specifically, we discuss the absence of a routine calibration procedure, the need for a robust analysis workflow to manage large datasets, and the implications of unassigned peaks, which we view as potential benefit rather than drawbacks. Furthermore, we note that the lack of automated $MS^2$ fragmentations remains a limitation, as these are currently performed manually, requiring pauses in online data acquisition. Additionally, we emphasize the aerosol mass-driven nature of the technique and its impact on our ability to observe e.g. new particle formation events.

We believe these additions enhance our manuscript by providing a more comprehensive understanding of the challenges and future directions for our approach. Thank you once again for your constructive feedback, which has helped us improve our manuscript

In conclusion, this study highlights the feasibility and versatility of APCI-Orbitrap-MS for field deployments, providing molecular-level insights into atmospheric processes. By expanding its use in field studies, this technology could extend capabilities of widely used aerosol measurement devices, offering enhanced data quality and a deeper understanding of the qualitative aerosol composition. Additionally, we demonstrated the potential of the APCI Orbitrap MS for qualitative measurements by using reference standards. Further studies will be required to be able to put this approach into practice, due to remaining challenges of limited availability of reference standards and the transferability of laboratory based calibrations to ambient setups. However, a few limitations remain that should be addressed in future studies. First, although we demonstrated the potential of APCI-Orbitrap-MS for qualitative measurements, routine calibration procedures still need to be developed using feasible reference standards. Second, the analysis of large datasets requires intensive post-processing, often resulting in many unassigned peaks. While this is mainly due to compound restrictions and not necessarily a disadvantage, it highlights the need for improved data analysis workflows. Third, fragmentation experiments currently must be triggered manually, causing interruptions in online data acquisition. Lastly, this method is driven by aerosol mass concentrations, which currently restricts our ability to measure or observe e.g. new particle formation events. Addressing these challenges would further improve the applicability of high resolution- MS for field deployment.

- The field data is interesting but I feel that the power of the high resolution isn't really drawn out. For instance, could you show two compounds that have very similar mass that you could not separate at lower resolutions? Or discuss how the observed trends for the target compounds might be incorrect using a lower resolution instrument.
Response:

We appreciate your suggestion to further emphasize the advantages of high-resolution measurements.

In the paper, we aimed to illustrate the importance of high resolution, particularly in Figure 6 panels b and c. In these figures, we present the measured spectra of levoglucosan, which is one main target compounds of our study. We demonstrate how the spectra could look like at two different resolutions, R = 4,000 and R = 14,000. Specifically, at a resolution of R = 4,000 in negative mode, levoglucosan could be

correctly identified because it appeared as the highest peak. However, at the same resolution in positive mode, it would not be possible to distinguish levoglucosan since is within the shoulder of the neighboring peak $C_{10}H_{11}O_2^+$.

This comparative analysis clearly shows that at lower resolutions, the accurate identification of levoglucosan would be compromised due to overlapping peaks. We believe this supports our argument regarding the necessity for high-resolution data in obtaining reliable results for compounds with close mass values. In combination with Figure 5, it shows in our opinion a clear picture about the strengths of high resolution.

Minor comments

- Figure 1 – this is blurry and hard to read. Its also not clear – are the blue and red clouds supposed to represent gas phase species?

  Response:

  Thank you for your feedback regarding Figure 1. We see how the current figure may lead to confusion regarding the blue and red clouds, which were intended to visualize the gas-phase. To avoid confusion, we will change all cloud colors to gray in the revised version of the figure. Additionally, we have reworked the figure to improve its resolution and ensure that all elements are clearly visible and easy to read.

[Figure]

- Table 1 seems unnecessary as most information is given in the text.

  Response:

  Thank you for your observation regarding Table 1. We understand your perspective that some information overlaps with the text. However, given that this is a methods paper, we believe that providing a summarized table allows readers to easily reference key settings for reproduction without needing to go through the detailed text.

  Additionally, we have included the AGC target value in Table 1, which is not explicitly stated in the text. We hope that this added detail enhances the table's utility for readers looking to replicate our methods.

**Table 1: Instrumental settings for online APCI-Orbitrap-MS measurements in negative and positive ionization mode.**

| | Negative ionization mode | Positive ionization mode |
|---|---|---|
| Vaporizer temperature (°C) | 350 | 350 |
| Ion transfer tube temperature (°C) | 200 | 200 |
| Ion source gas flows (a.u.): sheath-, aux-, sweep gas | 0* | 0* |
| Discharge current (µA) | 3 | 3 |
| Scan range (*m/z*) | 80–650 | 80–650 |
| RF lens (%) | 50 | 50 |
| AGC Target | $1 \times 10^6$ Standard | $1 \times 10^6$ Standard |
| Internal mass calibration (*m/z*) | 255.2329 | 203.0855[†] |
| Injection time (ms) | 100 | 100 |
| Averaged microscans | 10 | 10 |

*Sheath-, aux- and sweep gas were 0 a.u., due to the disconnection of $N_2$ supply.

[†]Internal calibration using EASY-IC™

- Section 2.2: The concentration used here are very high and this is likely to lead to different chemistry than in the real atmosphere. The impact of this should be acknowledged.
  Response:
  Thank you for your important comment regarding the concentrations of α-pinene and ozone used in our experiments. We acknowledge that these concentrations are higher than typical ambient levels and could potentially lead to different aerosol chemistry compared to the ambient atmosphere. Our primary objective in this study was not to investigate atmospheric processes in detail, but rather to produce a sufficient amount of aerosol mass to cover expected ambient concentrations in SKI. To achieve this, we increased the precursor concentrations, resulting in α-pinene SOA concentrations of 306.6 µg m$^{-3}$. This ensure that ambient aerosol concentrations lie well within this range, as we measured up 127 µg m$^{-3}$ (ACSM organics) in SKI.
  In light of your comment, we propose adding a disclaimer in Section 2.2 to clarify that our focus was on producing significant aerosol mass rather than replicating specific atmospheric chemistry. Moreover, we would like to highlight that similar concentrations of ozone have been reported in larger-scale studies investigating SOA formation from α-pinene, such as the work by Huang et al. (2018): "α-Pinene secondary organic aerosol at low temperature: chemical composition and implications for particle viscosity" (Atmospheric Chemistry and Physics, 18(4), 2883-2898).

to 6,000 ppb, resulting in a mean SOA mass concentrations of approximately 300 µg m$^{-3}$. We want to note that these precursor concentrations do exceed expected ambient concentrations and therefore do not necessarily represent atmospheric chemical

processes. These concentrations, however, were necessary to produce sufficient aerosol mass. We recorded the resulting
175 particle formation using a scanning-mobility particle sizer (SMPS, TSI, model: 3938) with a soft X-ray neutralizer.

- Line 253: what does "intensity" mean here? Do you mean the number of compounds?
  Response:
  In this context 'intensity' describes the summed peak intensity of the respective compound class. We changed the wording in Section 3.2.1 and 3.2.2.

  At CR approximately 2,000 molecular formulas (confidence level 4) were unambiguously identified and subsequently classified into the following compound groups, listed in descending order of summed peak intensity of the respective compound class: ~1,000 CHO, ~390 CHON, ~480 unclassified formulas (others), ~70 CHOS and ~80 CHNOS compounds
  260 (Fig. S8a).

  as indicated by the continuous measurement in negative ion-mode over a period of seven days (Fig. 4a). Approximately 3,500
  310 molecular formulas were assigned into the following compound groups in descending order of summed peak intensity of the respective compound class: ~1,100 CHO, ~1,300 CHON, ~440 unclassified formulas (others), ~230 CHOS and ~330 CHNOS compounds (Fig. S8b).

- Line 257: (and further refer to) – seems like the rest of this is missing?
  Response:
  Changed accordingly.

  Figure 3a shows the diurnal variation of three monomers $C_6H_{10}O_5$, $C_8H_{12}O_6$ and $C_{10}H_{16}O_3$ (left y-axis) and two dimers $C_{17}H_{26}O_8$ and $C_{19}H_{30}O_5$ (right y-axis) in local time. A distinct diurnal cycle was observed for $C_{10}H_{16}O_3$ (confidence level 4), which we attributed to pinonic acid and its isomers (and is further refer to pinonic acid only) based on its temporal behavior. We observed high intensities of pinonic acid at night and in the morning, while afternoon levels fell below the instrumental noise threshold,
  265 which could be explained by increased temperatures over the day resulting in the partitioning from pinonic acid from particle-

- Line 259: Could reduction in pinonic acid during the day also be related to secondary chemistry during the day?
  Response:
  Thank you for your comment regarding the potential relationship between the reduction of pinonic acid and secondary chemistry during the day. We assume "reduction" in this context was not meant as a chemical reaction but rather as a decrease in intensity.
  While the decrease of pinonic acid intensity could indeed be associated with secondary chemistry, we currently lack direct evidence for that hypothesis. Our findings suggest that

gas-particle partitioning of the IVOC pinonic acid, which is primarily influenced by temperature variations.

We have added a brief statement in the manuscript acknowledging that secondary chemistry may still play a role, even though we cannot confirm it based on our current data.

which could be explained by increased temperatures over the day resulting in the partitioning from pinonic acid from particle-into gas-phase. During the day additional reactions of pinonic acid due to secondary chemistry however can not be ruled out.

265    No clear diurnal pattern was observed for $C_8H_{12}O_6$ (confidence level 2), which we attribute to MBTCA based on its temporal

- Line 266: Levoglucosan behaved similarly to MBTCA. This seems unusual given there very different sources. In the fragmentation figure 8, you compare MBTCA to the PAM-OFR data. What does the comparison with the ambient data look like? Are you sure this is the correct species?

  Response:

  We appreciate the comment regarding the behavior of levoglucosan in comparison to MBTCA. We acknowledge that stating they behave similarly over time is somewhat misleading, as they originate from very different sources. Our wording was an unfortunate choice, based on the fact that we described the temporal evolution of MBTCA more detailed earlier in our discussion, which we than used as a comparison. In reality, both MBTCA and levoglucosan exhibit trends consistent with the overall pattern of total organic aerosol, as can be seen by the ACSM organics measurements in Figure 3b. Additionally levoglucosan displays a distinctive event-based increase on July 9th.

  Regarding the comparison of fragmentation data in Figure 8, we did not conduct fragmentation experiments of MBTCA in the field. Instead, we conduct the MS² experiments later in conjunction with the PAM-OFR setup to illustrate the applicability of our findings.

  We hope this clarifies our approach and the relationship between these compounds. Changes in the wording were done accordingly.

270    were highly correlated with the organic species measured by the ACSM. MBTCA showed slightly higher intensities in the late morning, around noon, with peak intensities on the 08 July at 10:00 h. Coupled with a strong decrease in the afternoon into the night, before reaching its second peak in the morning of 09 July between 06:00 h and 09:00 h. Similarly, Tthe BB marker $C_6H_{10}O_5$ (confidence level 2), levoglucosan, followed the overall pattern of ACSM organics behaved similarly to MBTCA, aside from an event on the evening of 09 July. Levoglucosan and its isomers (mannosan and galactosan), are well-established

- Figure 3: The colours are quite hard to differentiate – two greens and two blues. The should be changed to make it easier to read.
  We changed the colors in Figure 3a to enhance clarity and ensure easier differentiation.
- Figure 3: Can factor analysis be done on the ACSM data or is the resolution not good enough? It would have been nice here to show that the points that doent correlate as well were related to a higher f43 or other hydrocarbon fragment ion.
  Response:

Thank you for your comment regarding the potential for factor analysis on the ACSM data. However, we would like to clarify that conducting a factor analysis on ACSM data is outside the scope of this manuscript. Additionally, the $m/z$ fragment ions $C_3H_7^+$ and $C_2H_3O^+$, both of which contribute to the signal at $m/z$ 43 cannot be resolved by the ACSM deployed during our campaign, making it challenging to draw definitive conclusions regarding their correlation with higher $f43$ values. We appreciate your understanding and hope this explanation addresses your concern.

- Line 304: You have a lot of unassigned m/z values. Do you have any suggestions for why these are not assigned?
Response:
We would like to clarify that while we do have many $m/z$ values in our spectra for which no chemical formula could be derived, we have chosen not to address these in this paper.
The compounds referenced in line 305, which we categorize as 'unclassified formulas (others),' do have assigned chemical formulas. However, they do not fit within the categories of CHO, CHON, CHOS, and CHNOS. The ~440 unclassified compounds mentioned include high-intensity compounds such as $C_{30}H_{48}$ and $C_{26}H_{52}$. Additionally, this category encompasses formulas generated by Orbitool that, while valid in terms of their calculation, lack logical chemical compositions and therefore do not fall within the defined compound classes we are discussing.
We appreciate your understanding of these complexities and the rationale behind our focus in this analysis.

- Figure 4: Is there pinonic acid data for this site? It would be helpful to compare this between the two sites. Also, I would suggest that C8H13O8N could have multiple monoterpene sources rather than just a-pinene. Additionally, I would like to see a zoomed in version of 4c to see the correlation when the f44 is high.
Response:
Our answer is divided into several parts to address each concern regarding Figure 4 individually.
In response to the first part of the question, we had initially planned to include pinonic acid data for the SKI site; however, it was not detected at that location. For the second part of the question, we agree that $C_8H_{13}O_8N$ may have multiple monoterpene sources rather than solely α-pinene, and we will revise the text to reflect this. Our initial wording was based on observations during PAM-OFR experiments using α-pinene.

325    laboratory PAM-OFR experiments its formation by α-pinene ozonolysis in the presence of NO$_x$, h-owever other monoterpenes are also possible as precursors. During the night the planetary boundary layer (PBL) height is lower leading to more stable conditions and potentially higher concentrations of pollution. At the same time NO$_2$ is not photolyzed but instead reacts with O$_3$ forming NO$_3$, a common nighttime oxidant.

Regarding the third part of the question about the correlation plot in Figure 4c, we acknowledge correlation in the lower mass and intensity ranges cannot clearly be seen. To address this, we will include a zoomed-in version in the supplemental information.

[Figure]

**Figure S10: Pearson correlation of $\sum EIC_{neg}$ and ACSM organics species in SKI, colored by their respective $f_{44}$ values. For better visualization separated $f_{44} < 0.14$ on the left and f $f_{44} \geq 0.14$ on the right.**

- Figure S8: I think the legend is incorrect. The "other" category is black not cyan.
  Changed it accordingly.

- Line 375: Please give % of peaks that could be resolved at a resolution of 10k.
  Response:
  We added the percentage of peaks resolved below 10k accordingly.

  > positive mode, with the narrowest peak separation of 0.88×10⁻³ at *m/z* 89. Overall, the Δ *m/z* between peaks per unit mass
  > spans a wide range. While in negative mode, 19.5 % to ~40 % of the peaks could be resolved below 10k, in positive mode,
  > only 7.7 % to 18.6 % were resolved during BB events and non-BB events, respectively
  > 385  Meanwhile ~70k is required for achieving sufficient FWHM peak separation across the full
  > spectrum, especially up to *m/z* 250 in negative mode and *m/z* 350 in positive mode. In our ambient measurements, a resolution

- Line 416: I would remind the readers here that the standards were introduced as a nebulised methanol/water solution.
  Change it accordingly.

  > the APCI-Orbitrap-MS for ambient measurements, we investigated different biogenic (B-OA), anthropogenic (A-OA) and
  > biomass burning organic aerosol (BB-OA) compounds introduced by using a nebulizer. The compounds were dissolved in a
  > methanol-water solution and measured in both negative [M−H]⁻ and positive ionization mode [M+H]⁺ (Fig. 7).

- Section 3.5: This section needs to have a more critical evaluation of the spectra obtained. Some of the comparisons are not great – what are the similarity or reverse fit values? For MBTCA, why is the ambient data not used. Also, why have you chosen these three compounds? Are they fairly unique or dominant masses or simply because they are well known tracer compounds? At present, the identification of the pinene derived SOA components is not very convincing.

Response:

Regarding the similarity and reverse fit values: We did not calculate similarity or reverse fit values (such as those from Compound Discoverer or Mzmine) for our dataset. This decision stems from the lack of clarity in how these programs account for e.g. missing fragments, additional fragments, or variations in relative fragment intensities. We were concerned that providing similarity and reverse fit values would therefore not be comparable to common stated values and could potentially lead to false interpretation of our data. We believe that the fragmentation patterns observed for $C_6H_{10}O_5$ and $C_{19}H_{28}O_7$ are quite definitive, despite the absence of prior peak separation, which is often a main limitation in online measurements. Just to put our measurements into perspective, during typical offline measurements, fragmentation typically occurs for a single compound, and even then, the resulting data can sometimes be ambiguous. We also acknowledge that the case of MBTCA demonstrates the challenges associated with online $MS^2$ experiments, where fragmentation patterns may not be as clear. As mentioned in line 490, we have also identified additional compounds, such as $C_9H_{15}O_5^-$ and $C_{12}H_{11}O_2^-$, that undergo fragmentation and are present in the analysis window.

Regarding the use of laboratory-based data for MBTCA, we quite frankly did not conduct the fragmentation experiments during the field campaigns, so we had to use PAM-OFR experiments data.

As for the compounds selected for discussion, we chose $C_6H_{10}O_5$, MBTCA and $C_{19}H_{28}O_7$, because they are discussed throughout the paper, notably in Figures 3, 4, and 7. Additionally, we selected the $C_{19}H_{28}O_7$ dimer specifically because it represents a significant finding, as we were able for the first time to measurement this compound in ambient air, supported by $MS^2$ experiments.

While our analysis provides a probable structural attribution, it is important to note that with identification level 2, we cannot definitively rule out other precursors beyond monoterpenes. Classical offline LC-MS remains critical for achieving level 1 identification and confirming structures, as this approach can meet the necessary retention time and data requirements that our current APCI-Orbitrap-MS setup does not fulfill.

We appreciate your insights and hope this response clarifies our approach and rationale.

which results from the ester cleavage into cis-pinic acid and $C_{10}H_{15}O_4^-$. Aside from this, the main fragments in ambient air were $m/z$ 349 ($C_{19}H_{25}O_6^-$, $[M-H-H_2O]^-$) and $m/z$ 323 ($C_{18}H_{27}O_5^-$, $[M-H-CO_2]^-$).

It is noteworthy, that with identification level 2 only a probable structure can be attributed, which means that other precursors besides monoterpenes cannot be entirely ruled out. For this, identification level 1 (confirmed structure) would be necessary, however this cannot be achieved with the APCI-Orbitrap-MS setup, as it does not have a retention time and therefore cannot meet the minimum data requirement. For this additional LC-MS measurements would be required. Nevertheless, $MS^2$ experiments offer a more specific evaluation of single ions, the lack of upstream separation and currently no automatized

- Line 450: What is the m/z isolation window used for MS2 and are there any other peaks found within this window.

Response:

In response to the inquiry about the $m/z$ isolation window used for $MS^2$, we used an isolation window of ± 1 $m/z$ for all compounds. In retrospect, a smaller window might have been preferable, but we maintained this setting for consistency with ambient data. We will add this information additionally in the figure caption to have this information more present.

Regarding other peaks found within this window, as mentioned in line 475, there are other ions, such as $C_9H_{15}O_5^-$ and $C_{12}H_{11}O_2^-$, which also undergo fragmentation within the MBTCA isolation window of $\pm 1$ $m/z$. Additionally, in line 464 and following, we noted some high-intensity fragments recorded during field measurements, which could potentially be attributed to the molecular ion $C_5H_7O_3^-$, which lies within 0.03 amu of $C_6H_9O_5^-$ in the isolation window.

Figure 8: Mirror spectra for online MS² experiments of reference measurements or spectral libraries (top spectra respectively) and complex ambient or laboratory aerosol mixtures (bottom spectra respectively) for a) $C_6H_{10}O_5$, b) $C_8H_{12}O_6$ and c) $C_{19}H_{28}O_7$. The isolation window was kept constant at $\pm 1$ $m/z$.